# TomoTwin: generalized 3D localization of macromolecules in cryo-electron tomograms with structural data mining

Gavin Rice[1,2], Thorsten Wagner [1,2], Markus Stabrin [1], Oleg Sitsel [1], Daniel Prumbaum[1] & Stefan Raunser [1] ✉

Cryogenic-electron tomography enables the visualization of cellular environments in extreme detail, however, tools to analyze the full amount of information contained within these densely packed volumes are still needed. Detailed analysis of macromolecules through subtomogram averaging requires particles to first be localized within the tomogram volume, a task complicated by several factors including a low signal to noise ratio and crowding of the cellular space. Available methods for this task suffer either from being error prone or requiring manual annotation of training data. To assist in this crucial particle picking step, we present TomoTwin: an open source general picking model for cryogenic-electron tomograms based on deep metric learning. By embedding tomograms in an information-rich, high-dimensional space that separates macromolecules according to their three-dimensional structure, TomoTwin allows users to identify proteins in tomograms de novo without manually creating training data or retraining the network to locate new proteins.

Cryogenic-electron tomography (cryo-ET) has emerged as a landmark technique for the visualization of macromolecules within their native cellular environment[1–7]. Advances in high-pressure freezing and focused ion beam milling at cryogenic temperatures now allow for the routine preparation of thin (less than 200 nm) lamellae from cells or even small organisms[8–10]. Cryo-ET offers a unique opportunity to capture cellular processes in three dimensions and in unprecedented detail, and subsequent analysis of specific macromolecules from tomograms through subtomogram averaging (STA) allows for in-depth structural determination of macromolecular complexes in situ[11–14]. Particularly when complemented by recent advances in structure prediction such as alphafold2, STA forms a powerful crossbridge between protein biochemistry and cellular proteomics[15–17]. To perform STA, however, particles of a macromolecule of interest must first be located within the tomograms, a task complicated by the three-dimensional (3D) nature of these data.

The accurate localization of macromolecules inside cryo-electron tomograms is a well-recognized barrier for studying cellular life at the mesoscopic level[18]. This has led to the development of several deep learning-based tools often leveraging popular 3D-Unet convolutional neural network (CNN) architectures[19–21]. None of these approaches, however, have been able to demonstrate generalization, meaning that for each protein of interest, users must first manually annotate hundreds to thousands of particles in tomograms and train the neural network to identify that protein. Not only is this incompatible with the future directions of automated tomogram reconstruction and STA, but for most of the proteome, manually annotating sufficient training data from experimental tomograms to train the network is not possible. As a result, these deep learning tools cannot currently assist in answering many of the outstanding biological questions for which cryo-ET presents vast potential. Owing to this issue of usability, template matching[22,23] is still highly used in cryo-ET processing workflows, especially those that place an emphasis on throughput[24], although this technique suffers notably by comparison in terms of picking accuracy, often limiting the overall effectiveness of STA.

[1]Department of Structural Biochemistry, Max Planck Institute of Molecular Physiology, Dortmund, Germany. [2]These authors contributed equally: Gavin Rice, Thorsten Wagner. ✉e-mail: stefan.raunser@mpi-dortmund.mpg.de

To date, no methods have demonstrated the accuracy of deep learning-based picking with an unexceptionable level of usability for cryo-ET. One method to retain this accuracy while circumventing the requirement of manually annotating training data for each protein of interest is to train a model to learn a generalized representation of a 3D molecular shape that then can differentiate between macromolecules based on their structure. Such general models have seen a high uptake in two-dimensional (2D) particle picking for single particle cryo-electron microscopy (cryo-EM) analysis[25-28] although the translation of these methods to tomograms is still lacking due to the additional challenges posed by 3D tomography data.

Particularly well suited to this challenging case of generalization is deep metric learning in which data are encoded as a high-dimensional representation, called an embedding[29,30]. During training, the model is penalized for placing data from different classes near to one another and rewarded for placing data from the same class close together in the embedding space[31]. Therefore, over the training process the model learns to cluster each class in a distinct region of the embedding space where similar classes are placed closer together and dissimilar ones further apart. In some cases, the embeddings of a dataset are sufficiently ordered to allow for de novo identification of classes in the embedding space[31]. By understanding similarity relationships, deep metric learning models have demonstrated an acute generalizability, being able to place new classes of data in the embedding space according to their similarity to known classes without requiring retraining[31-33].

Here we present TomoTwin, a generalized particle picking model and deep metric learning toolkit for structural data mining of cryo-electron tomograms. We supply two workflows for macromolecular localization with TomoTwin, a reference-based workflow in which a single molecule is picked for each protein of interest and used as a target, and a de novo clustering workflow where macromolecular structures of interest are identified on a 2D manifold. Trained on a diverse set of simulated tomograms, the picking model of TomoTwin can locate new proteins with high accuracy in not only simulated data, but in experimental tomograms as well. By removing the steps of annotating training data and retraining a picking model for each protein, TomoTwin combines the accuracy of deep learning-based particle picking with a high degree of usability and allows for the simultaneous picking of several proteins of interest in each tomogram.

## Overview of functions, build and philosophy behind TomoTwin

The machine learning backbone of TomoTwin is built on the principle of learning generalized representations of 3D shapes in tomograms (Extended Data Fig. 1b,c). Trained with deep metric learning, the 3D CNN is able to locate not only macromolecules from the training set, but generalize to new macromolecules as well, allowing TomoTwin to retain the high fidelity of deep learning-based picking while avoiding the burden of requiring retraining for each protein of interest. The general model embeds tomograms by sampling overlapping subvolumes and embedding them according to the similarity of their macromolecular contents (Extended Data Fig. 1d). Once a tomogram is embedded, particles of each macromolecule can be picked by identifying their associated region in the embedding space. This can be done either by identifying a single example of each protein of interest in a tomogram and using them to mark the region of the space where they are embedded (reference-based workflow), or by approximating the tomogram embeddings onto a 2D manifold where clusters for each macromolecule can be identified by eye (clustering workflow) (Fig. 1a,b). Once the embeddings containing a protein of interest are identified, they must be mapped back to the tomogram where overlapping picks of the same molecule can be consolidated into one centralized pick per molecule (Fig. 1c). Finally, TomoTwin allows users to interactively filter the picked particles for each macromolecule of interest based on the particle size and the distance between each particle and the target

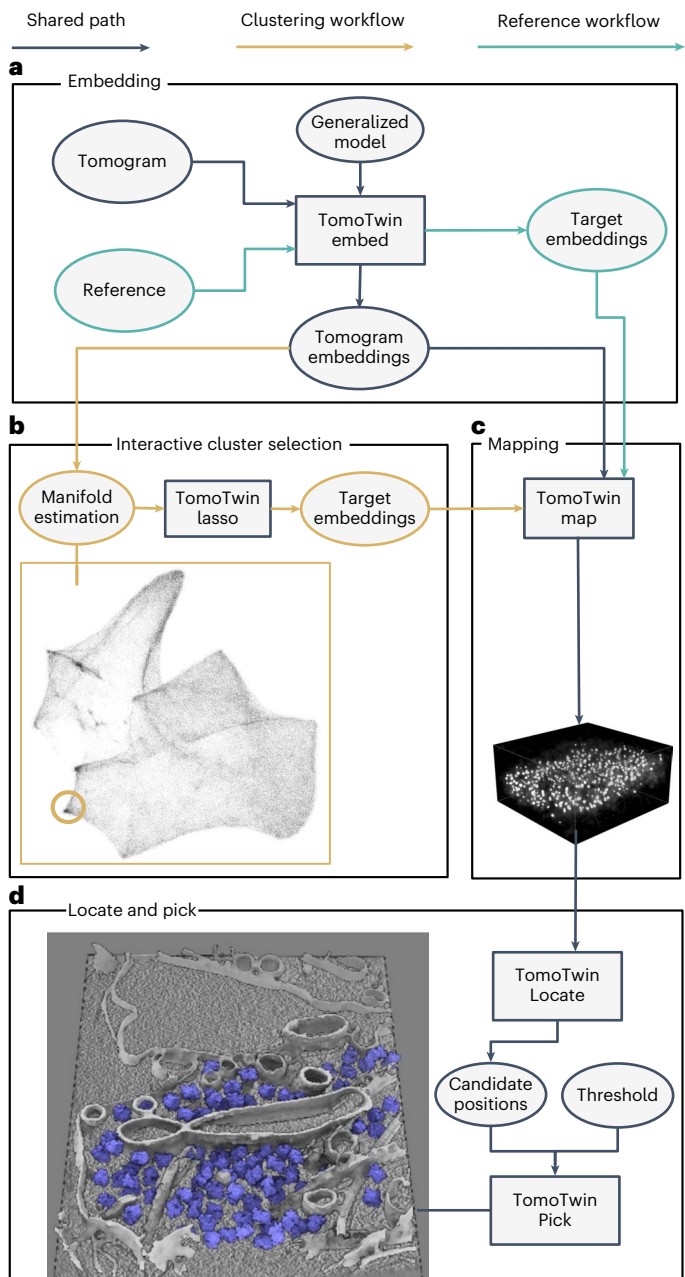

**Fig. 1 | TomoTwin identifies and localizes particles by a clustering or a reference-based workflow. a**, The first step in using TomoTwin is to embed the tomogram with the pretrained model. Optionally, references can be selected and embedded as well to create target embeddings. **b**, For the clustering workflow the tomogram embeddings are projected on a 2D manifold and an interactive lasso tool is used to select clusters of interest to generate target embeddings. **c**, The distance matrix between each target embedding and the embeddings of the tomogram is calculated. **d**, All local maxima are located with TomoTwin Locate and are used to pick final coordinates for each protein of interest using TomoTwin Pick with confidence and size thresholding.

embedding for that macromolecule in the embedding space (Fig. 1d and Extended Data Fig. 1a).

## Two workflows to locate macromolecules in tomograms

TomoTwin embeds tomograms in a high-dimensional space where subvolumes of each macromolecule are located in a distinct region of

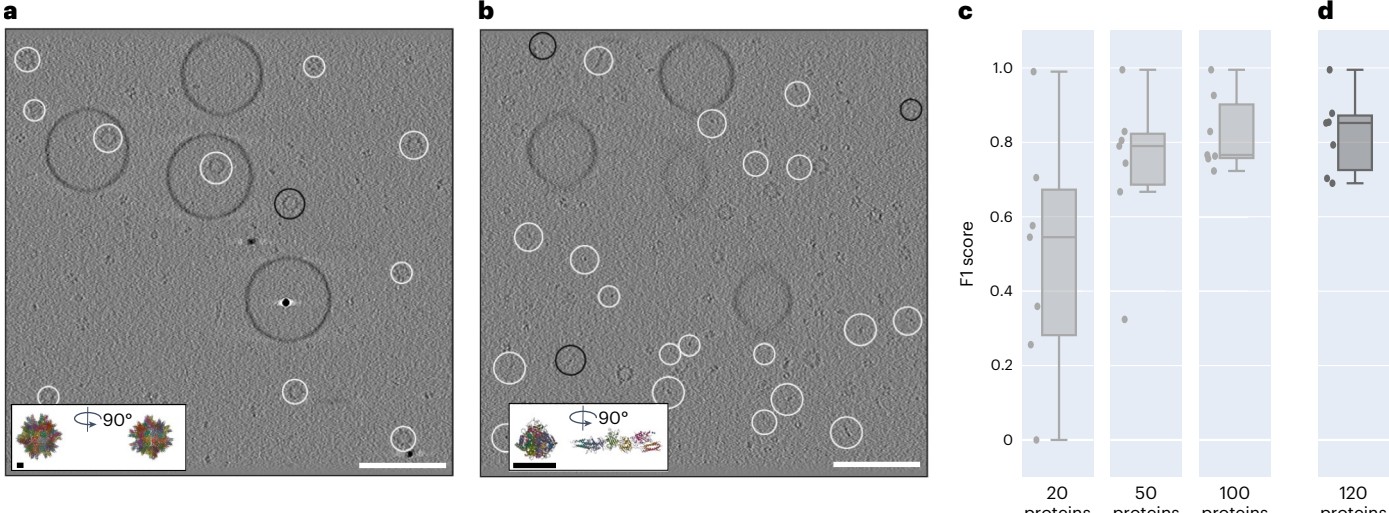

**Fig. 2 | The generalized model of TomoTwin locates novel proteins accurately. a**, True-positive selected particles (white) and false negative (black) of the largest protein (PDB ID 2DF7) (896 kDa) in the generalization tomogram. **b**, The smallest protein (PDB ID 1FZG) (142 kDa) in the generalization tomogram. The F1 scores are 0.99 and 0.88 for largest and smallest protein, respectively. **c**, When increasing the number of proteins used during training, the mean F1 score for each protein (*n* = 7) in the generalization tomogram increased as well. The mean

F1 scores are 0.49, 0.73 and 0.82 for a model trained on 20, 50 and 100 proteins, respectively. **d**, The model trained on the full training set of 120 proteins reached a mean F1 score of 0.82 but has the highest median F1 score of 0.85. Each box in **c** and **d** extends from the first (Q1) to the third quartile (Q3). The median is marked by a line inside the box. Whisker lines correspond to box edges ±1.5 times interquartile range. White scale bars, 100 nm; black scale bars, 5 nm.

the space. To identify the location of a macromolecule in the embedding space, we provide the user with two workflows: a reference-based workflow and a clustering workflow. Each workflow picks particles with high accuracy, but the reference-based approach begins with identifying an example of the protein of interest in the tomogram and mapping this to the embedding space whereas the clustering workflow begins with identifying a region of the embedding space and mapping this to the tomogram. Which workflow is most suitable for any given application depends on how easily the protein(s) of interest can be identified in the tomogram versus the embeddings. If an example particle of a protein can be identified in a tomogram, the reference-based workflow offers a streamlined picking approach. Conversely, the clustering workflow offers advantages for exploring the macromolecular contents of a tomogram without requiring a priori knowledge. Both workflows share the common first step of using the embedding function of TomoTwin to generate a high-dimensional embedding of the entire volume of the tomogram (Fig. 1a).

In the reference-based workflow, a single molecule of each protein of interest in a tomogram is embedded to generate a target embedding for that protein. In the clustering workflow, the tomogram embeddings are approximated onto a 2D manifold. This 2D manifold can then be directly used to outline one or more clusters of interest using an interactive tool. The mean embedding of the enclosed subvolumes of each cluster is then used as a target embedding in lieu of a reference (Fig. 1b). The Map function of TomoTwin takes as input the tomogram embeddings and target embedding(s) and calculates the distance between the target(s) and each subvolume in the embeddings. These distances are mapped to the subvolume positions, constructing a map of proposed particle locations within the tomogram for each protein of interest (Fig. 1c). The Locate function uses this map to localize peaks of high similarity and generate candidate particle positions. Finally, the Pick function of TomoTwin in tandem with the graphical user interface uses these candidate positions along with adjustable size and similarity thresholds to pick particles in the tomogram producing a coordinate file for each protein of interest suitable for STA or other analysis (Fig. 1d).

## Training of the general picking model

TomoTwin is trained using deep metric learning on triplets of subvolumes from simulated tomograms. The triplets are constructed as sets of three subvolumes each containing a particle, two containing the same protein and one a different protein, called the anchor, positive, and negative respectively. A set of 120 structurally dissimilar proteins procured from the Protein Data Bank (PDB) ranging in size from 30 kDa to 2.7 mDa were used to simulate 84 tomograms containing a total of 120,000 particles (Extended Data Fig. 2). During training, batches of subvolumes are embedded by the 3D CNN that transforms each 37 × 37 × 37 subvolume to a 32-length feature vector located on a high-dimensional embedding manifold molded to the surface of a 32D hypersphere (Extended Data Fig. 1b). These feature vectors are then used for metric learning.

Through training, TomoTwin learns to place each macromolecule within a distinct region of the embedding space, where more structurally similar macromolecules are placed closer together and dissimilar ones further apart (Extended Data Fig. 1d). By training on a large, diverse set of 3D macromolecular shapes and sizes, TomoTwin learned a generalized representation of 3D macromolecular shapes that it leverages to place new macromolecules in the embedding space relative to their structural similarity to known proteins without requiring retraining.

## The picking model generalizes across protein shape and size

Because a priori information on the ground-truth locations of all molecules in a tomogram is not possible to obtain for experimental data, we first assessed the picking performance of the trained model on simulated tomograms containing proteins from the training set.

The median F1 picking score across all validation tomograms was 0.88 with a range from 0.76 to 0.98 (Extended Data Fig. 3a). Across all proteins in the training set ranging from 30 kDa to 2.7 mDa, the median validation F1 picking score is 0.92 (Extended Data Fig. 3b). In rare cases, outlier scores were observed where specific proteins could not be picked across a range of sizes (Extended Data Fig. 3c). Closer inspection

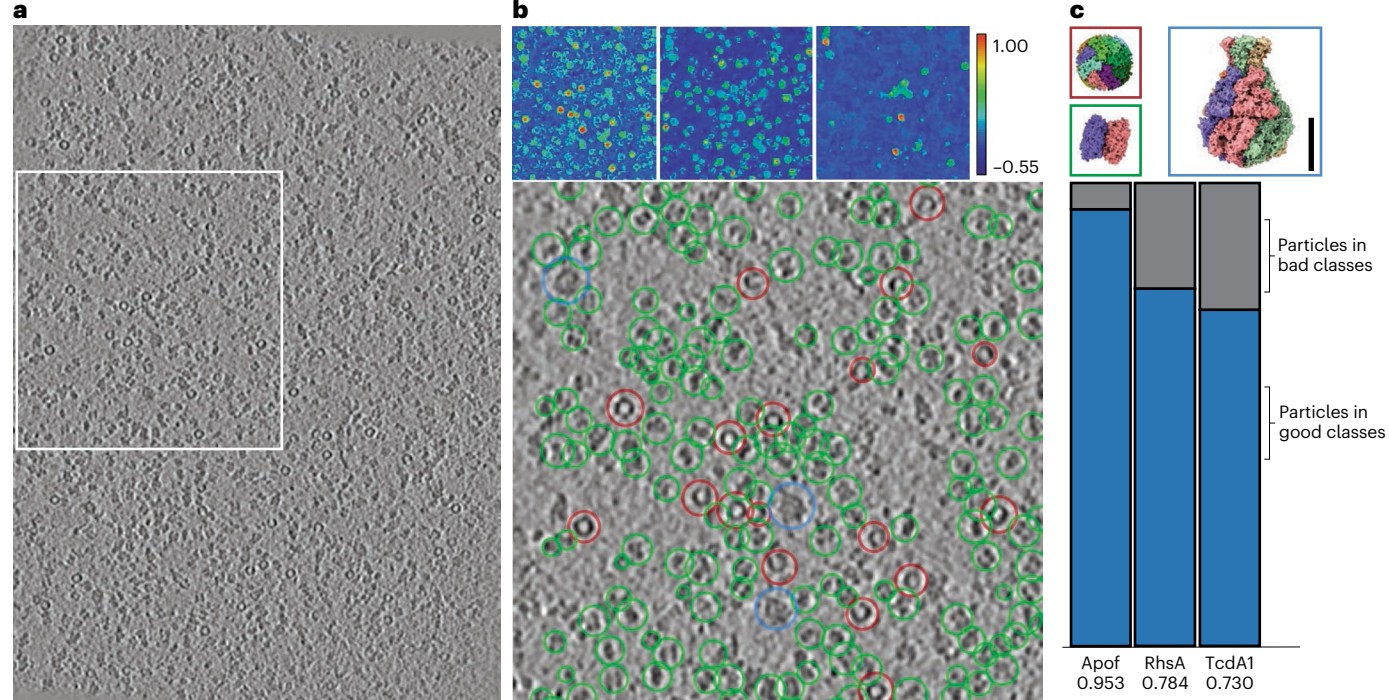

**Fig. 3 | TomoTwin accurately localizes multiple proteins simultaneously in crowded tomograms. a**, Representative slice of a tomogram containing a mixture of apoferritin, RhsA and TcdA1. Scale bar, 100 nm. **b**, Cosine similarity heatmap and representative picking for apoferritin (red), RhsA (green) and TcdA1 (blue), respectively. Total subvolumes picked were: apoferritin, 848; RhsA, 2196; and TcdA1, 122. Scale bar, 50 nm. **c**, Protein structures of apoferritin (PDB ID 1DAT), RhsA (PDB ID 7Q97) and TcdA1 (PDB ID 6L7E) and the ratio of picked subvolumes contained in selected good and bad 2D classes. Scale bar, 100 nm. Supplementary Video 1 is a video going through the tomogram in the *z* direction with the individual picks for TcdA1, apoferritin and RhsA highlighted.

of these outliers revealed that in the simulated tomograms, each of these proteins display a particularly weak signal when compared to proteins of similar size (Extended Data Fig. 3d). In these cases, the proteins display a shape that is not recovered well during tomogram reconstruction by weighted back projection. Despite this, picking on the validation tomograms demonstrated high accuracy for proteins across a wide array of shapes and sizes.

## TomoTwin generalizes to unseen proteins

To assess the generalization of the general picking model to particles that were not in the training dataset, we measured the picking performance on a simulated tomogram, called the generalization tomogram that contains proteins not included in the training set, with the reference-based workflow. When trained on a set of 120 dissimilar proteins (Extended Data Fig. 2), the resulting model was able to locate all seven proteins accurately with a median F1 score of 0.82 despite a lack of previous training on these proteins (Fig. 2d). To measure the effect of training set size on generalization accuracy we performed this analysis on picking models trained on 20, 50, 100 and 120 proteins where we observed a logarithmic increase in generalization accuracy with the number of proteins in the training set (Fig. 2c). This high accuracy in locating novel proteins indicates a high generalization capability of TomoTwin, a feat no other deep learning picking method for cryo-ET has reported so far.

To quantitatively measure the effect of increased particle density on picking quality, we simulated an additional generalization tomogram in which the number of particles per protein in the tomogram is increased by fivefold to replicate a highly crowded environment (Extended Data Fig. 4a). In this densely packed tomogram, we observe an overall mean F1 picking score of 0.82 indicating that while particle picking in dense environments containing many proteins poses an additional challenge, the picking performance of the TomoTwin general model remains unequivocal.

## TomoTwin picks proteins accurately in experimental tomograms

As TomoTwin is trained entirely on simulated data, it is paramount to investigate its ability to pick proteins of interest in experimental tomograms. To evaluate this, we tested the picking accuracy of TomoTwin on several experimental datasets.

First, cryo-ET was performed on a sample containing a mixture of three proteins, apoferritin[34], the Type VI secretion effector RhsA from *Pseudomonas protegens*[35] and the Tc toxin A component TcdA1 from *Photorhabdus luminescens*[36] as well as liposomes (DOPC/POPC) (Fig. 3a). This mixture was chosen to create a complex, crowded environment in vitro that may confound picking accuracy. Ten reconstructed tomograms were picked for apoferritin, RhsA and TcdA1 using the general model. In each case, the reference-based workflow was used in which a target embedding was created by picking a single example of each protein in one tomogram. The target embedding of each protein was then applied for picking across all tomograms in the dataset. Direct visualization of the similarity maps and picks reveals that TomoTwin identified each protein in the mixture with minimal confounding from the crowded environment, contaminants, degraded particles or non-target proteins to achieve high-fidelity picking (Fig. 3b). Additionally, the percentage of particles in good 2D classes was measured as a relative indicator of the precision (Fig. 3c and Extended Data Fig. 5c–e). This is further confirmed by manual calculation of recall and precision for TcdA1 (recall 0.81, precision 1.0) and apoferritin (recall 0.91, precision 1.0) by counting picked and missed particles (Extended Data Fig. 5a,b).

This picking result affirms several aspects of the generalizability of TomoTwin for particle picking in tomograms. Foremost, that TomoTwin generalizes to new proteins without requiring retraining as none of these proteins were included in the training set. Of equal importance, the high-fidelity picking indicates that TomoTwin, trained on simulated tomograms, can also be applied to experimental tomograms.

One of the principal advantages of cryo-ET is the ability to directly visualize proteins in their native cellular environments. Owing to crowding of the cellular space and the poor contrast caused by thick specimens however, particle localization within a cellular environment presents a substantial challenge. To assess its ability to locate particles in cellular tomograms, we applied TomoTwin to a dataset of tomograms containing *Mycoplasma pneumoniae*[37] (EMPIAR 10499) (Extended Data Fig. 6a). Using the TomoTwin general model, we picked 70S ribosomes in 65 tomograms with the reference-based workflow in which a single ribosome was identified in a tomogram and used to generate a target embedding that was then applied to pick the entire dataset (Extended Data Fig. 6b,c). To visualize the results, we extracted pseudo-subtomograms[38] and performed 3D classification using a 70S ribosome cryo-EM structure (EMD-11650), lowpass filtered to 30 Å as a reference. As all 3D classes resemble ribosomes refined to roughly 15 Å, it clearly indicates that TomoTwin also picks highly accurately in cellular tomograms (Extended Data Fig. 6d).

## TomoTwin establishes new standards of accuracy and usability

To establish how TomoTwin compares with available methods for particle picking in cryo-ET, we directly measured the picking performance first against template matching and subsequently against nongeneralizing machine learning workflows.

Several software are available for template matching in cryo-ET including EMAN2 (ref. 39), Dynamo[40] and PyTom[41], from which we selected EMAN2 as a representative case. When applied to the validation dataset of 120 proteins ranging in size from 30 kDa to 2.7 mDa, TomoTwin demonstrated superior picking performance compared to EMAN2 as measured by the F1 accuracy score as well as a greater consistency in picking accuracy across the entire range of proteins (Extended Data Fig. 7a). Moreover, this advantage was carried over to picking performance on the generalization tomogram (Extended Data Fig. 7b).

Directly comparing TomoTwin against nongeneralizing machine learning picking workflows is more challenging as the main distinction comes at the level of usability rather than statistical picking accuracy. The advantage of nongeneralizing deep learning approaches, such as DeepFinder[19], over template matching has been previously demonstrated in the ability to achieve a low-resolution reconstruction of the enzyme ribulose-1,5-biphosphase carboxylase-oxygenase (RuBisCO) with fewer particles than were needed when picking with template matching[19]. To achieve this, however, first template matching was used to pick RuBisCO in tomograms, then manual curation of the picks to reach more than 175,000 annotated particles to train and validate a model for picking the enzyme in the remaining tomograms. While producing a high accuracy overall, a combination of template matching and manual picking was used to generate training data and the training step alone in this workflow required 35 hours. This sample is particularly suitable for assessing the performance of particle picking in a crowded cellular environment due to the unconventional phenomenon of RuBisCO packing into a densely packed protein matrix inside the pyrenoid. When applied to a tomogram from the same dataset (EMPIAR 10694), TomoTwin was able to localize RuBisCO with a recall of 0.8 (Extended Data Fig. 8) based on a single reference. These picks proved sufficient for direct analysis by STA surpassing the previously reported resolution while maintaining a highly efficient workflow (Fig. 4). Comparing the map of RuBisCO from TomoTwin raw picked particles with the original (EMD-3694)[42] shows that the reconstruction from TomoTwin picked particles is of a comparable if not superior quality and thus TomoTwin picking did not limit the resolution that can be achieved by STA (Extended Data Fig. 9e). Furthermore, as a result of not requiring the user to manually annotate training data or retrain the network to pick these particles, TomoTwin was able to generate

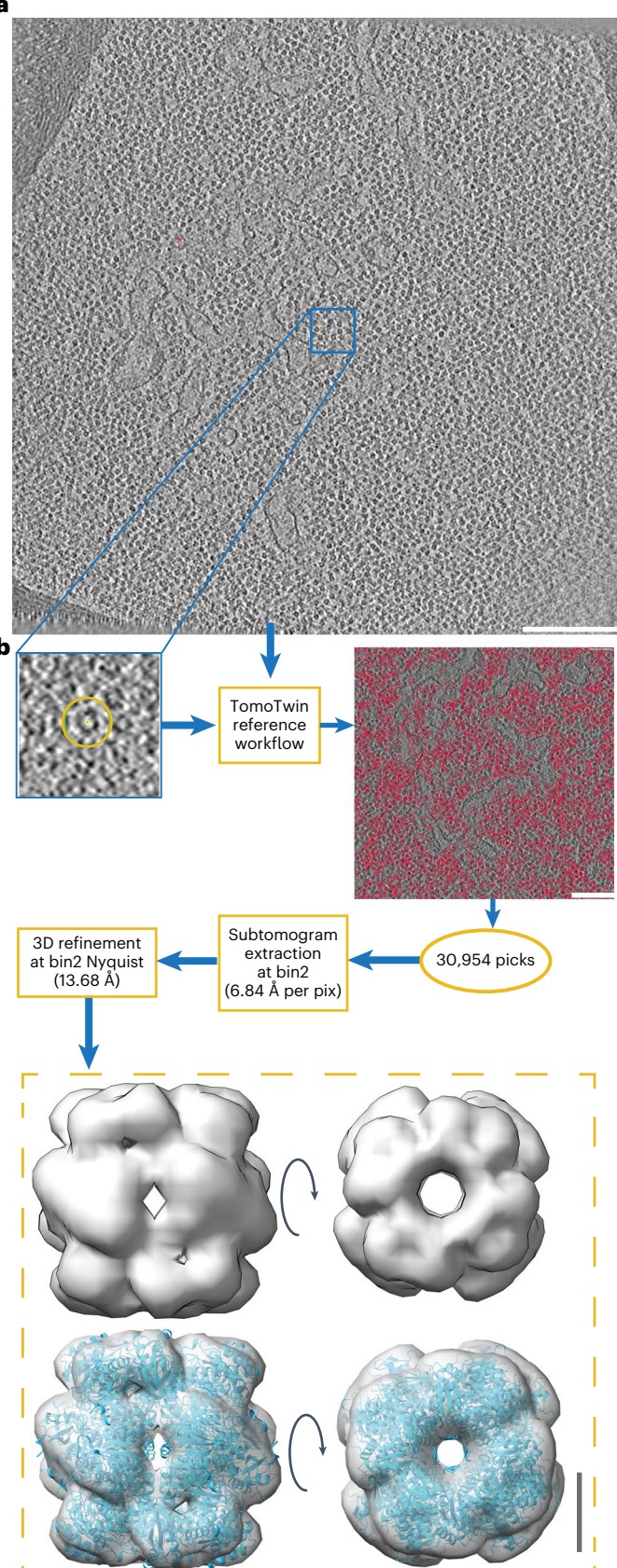

**Fig. 4 | TomoTwin matches the quality of supervised deep learning approaches without any training required. a**, Slice of a tomogram containing a *C. reinhardtii* pyrenoid. Scale bar, 200 nm. **b**, Picking and processing pipeline. Picking scale bar, 100 nm; 3D refinement scale bar, 4 nm.

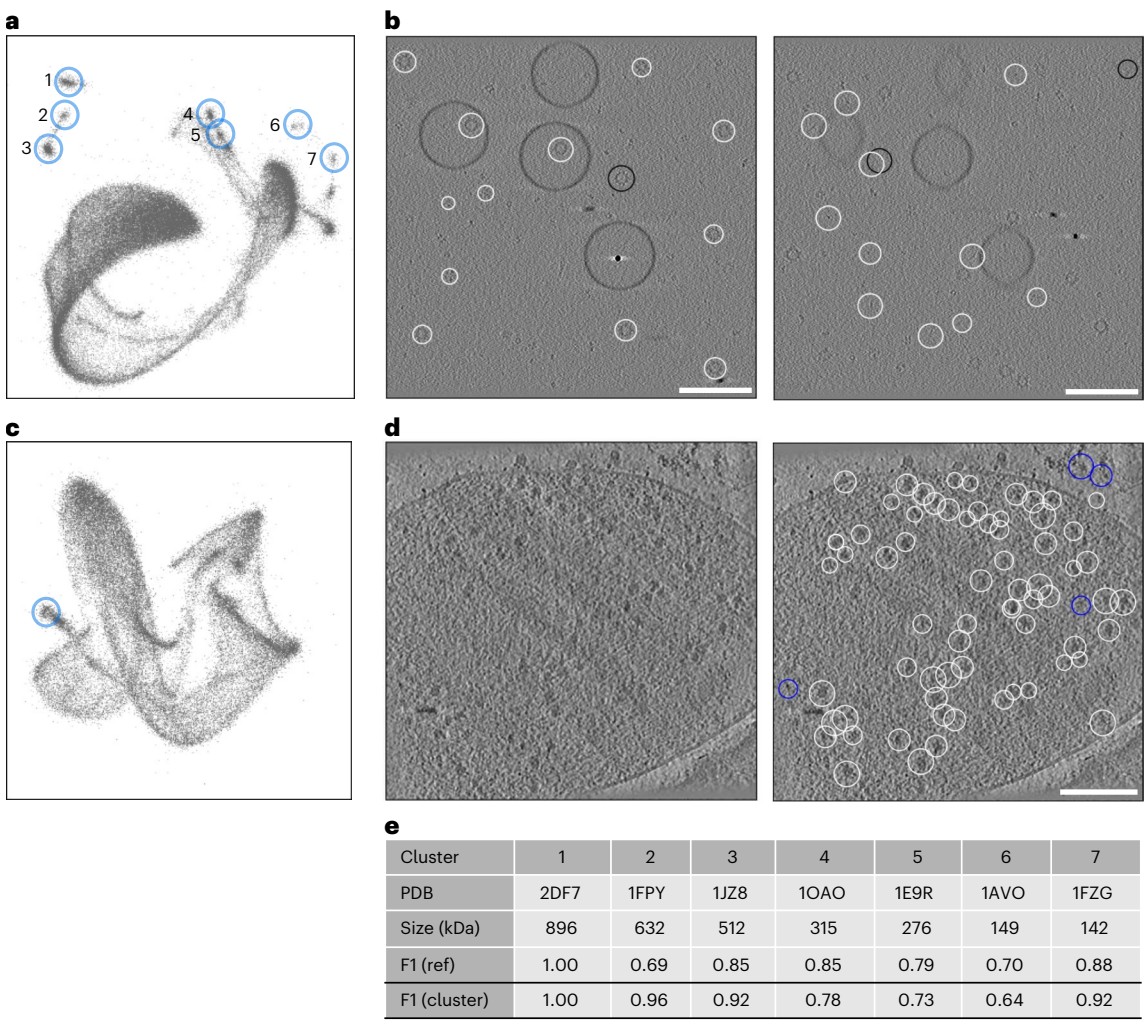

| Cluster | 1 | 2 | 3 | 4 | 5 | 6 | 7 |
|---|---|---|---|---|---|---|---|
| PDB | 2DF7 | 1FPY | 1JZ8 | 1OAO | 1E9R | 1AVO | 1FZG |
| Size (kDa) | 896 | 632 | 512 | 315 | 276 | 149 | 142 |
| F1 (ref) | 1.00 | 0.69 | 0.85 | 0.85 | 0.79 | 0.70 | 0.88 |
| F1 (cluster) | 1.00 | 0.96 | 0.92 | 0.78 | 0.73 | 0.64 | 0.92 |

**Fig. 5 | TomoTwin enables structural data mining on the embedding manifold. a**, Highlighted clusters of all seven proteins on the generalization tomogram 2D manifold approximation. **b**, Respective particle locations from cluster 3 (left) and 5 (right) that correspond to the proteins with PDB ID 2DF7 and 1FZG, respectively. The true-positive picks are white and false negatives are black. In both cases there were no false-positive selections. **c**, 2D manifold approximation of the embedding space of a tomogram containing *M. pneumoniae* (EMPIAR 10499). The manual selected cluster is highlighted

that corresponds to the 70S ribosome. **d**, Using the cluster center for picking identified all ribosomes previously selected by the reference-based picking (white) with a few reference-only selections (blue). **e**, F1 scores for the individual clusters in comparison with the F1 scores for reference-based picking. On average, the clustering performed slightly better (0.84 versus 0.82 mean F1 score). More detailed metrics can be found in Extended Data Fig. 7f,g. Scale bar, 100 nm.

these accurate picking coordinates with a total working time of under 1 hour. When applied to the same tomogram, the clustering workflow achieved similar picking results (Extended Data Fig. 8).

## TomoTwin generalizes across a variety of experimental setups

In addition to generalizing to pick new proteins, a highly usable picking tool must also generalize across a variety of common experimental practices in cryo-ET.

While trained solely on simulated tomograms with a total dose of 150 e/Å$^2$, collected from tilt series ranging from −60° to 60° in 2° increments, the aforementioned picking examples on experimental tomograms demonstrate several additional degrees of generalization including tilt range, tilt step, total dose and detector (Extended Data Fig. 7e).

To further analyze the effect of varying experimental parameters quantitatively, several congruent simulated generalization tomograms were created wherein the tilt range and total dose were varied to

account for a range of possible cryo-ET experimental setups (Extended Data Fig. 7). To control for possible bias from the reference particle used for picking, five reference particles were used for each protein and the reference that returned the most consistent picking result across the parameters was reported.

Decreasing the tilt range increases the effects of the missing wedge artifact on the reconstructed tomogram resulting in a stronger deformation of each reconstructed particle[43]. This effect is particularly pronounced on long, thin particles such as carbon monoxide dehydrogenase (PDB 1OAO) making accurate particle picking substantially more challenging. Overall, we observe a 5.4 and 10.3% decrease in mean F1 picking performance when the tilt range is restricted from −60°60° to −50°50° and −40°40°, respectively (Extended Data Fig. 7c).

Similarly, decreasing the total electron dose directly reduces the protein signal resulting in a tomogram with a reduced signal to noise ratio. We observe a 2.3 and 8.6% decrease in F1 picking performance when the total dose is restricted from 150 to 135 and 120 e/Å$^2$, respectively (Extended Data Fig. 7d).

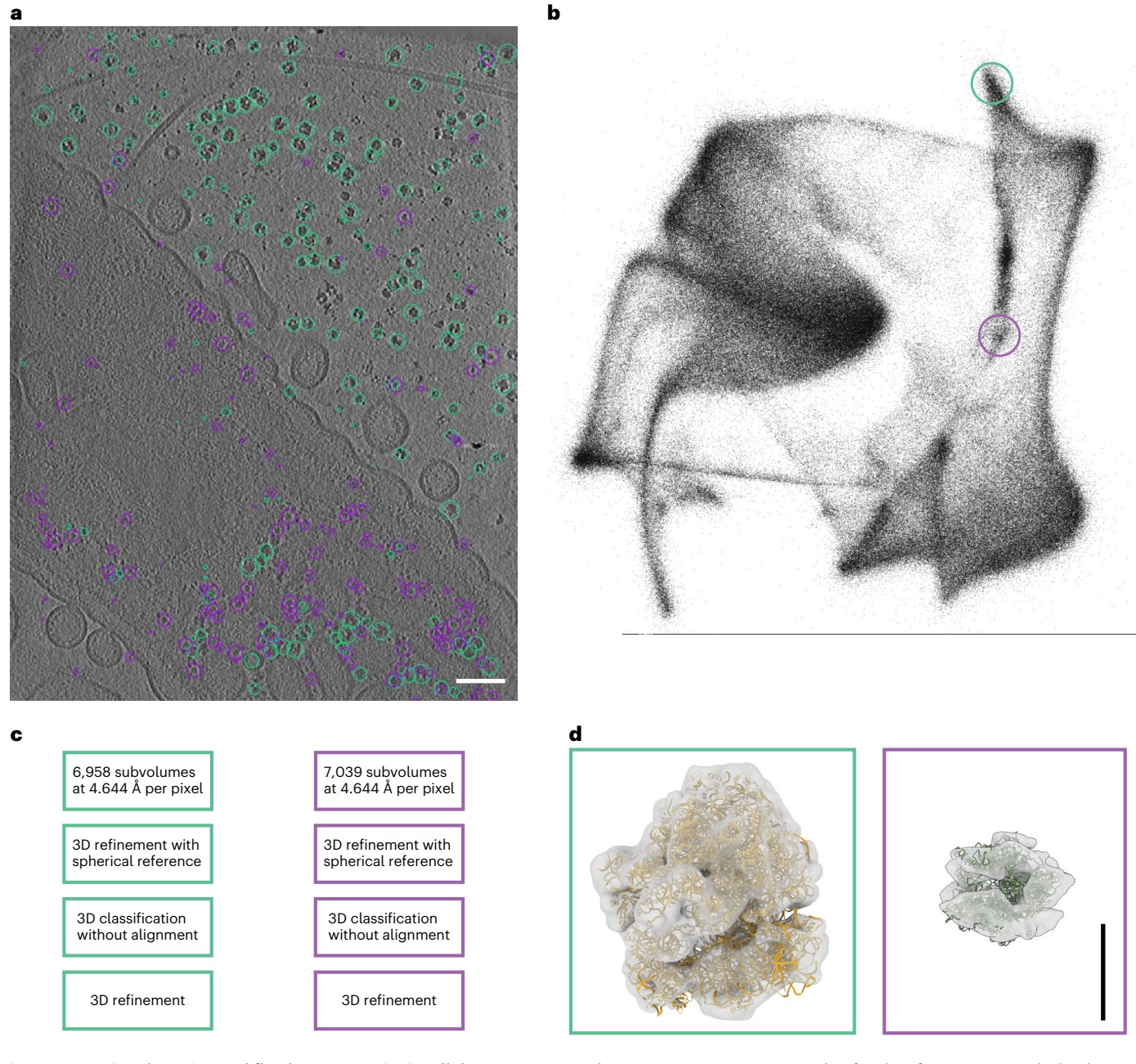

**Fig. 6 | TomoTwin's clustering workflow locates proteins in cellular tomograms de novo. a**, Slice through a tomogram containing a *Y. entomophaga* cell. Coordinates picked from two separate clusters highlighted in green and purple. Scale bar, 100 nm. **b**, TomoTwin representation map for the tomogram shown in **a**. Clusters corresponding to picks are highlighted in the same color scheme as **a**. **c**, STA processing pipeline for identifying proteins picked with the clustering workflow. **d**, 3D refined maps from STA analysis fit with a bacterial ribosome (left) (PDB ID 7K00) and a bacterial RNA polymerase (right) (PDB ID 1HQM). Scale bar, 10 nm.

Overall, while a decrease in picking performance with the increase in corruption or lack of protein signal in tomograms is expected, TomoTwin leverages the high degree of adaptability rooted in its deep metric learning backbone to achieve accurate particle picking.

## Structural data mining on the embedding manifold

The embedding feature of TomoTwin maps tomograms as a series of embeddings organized according to their macromolecular contents. Such organized embeddings present ideal candidates for structural data mining. These embeddings can be directly visualized by approximation on a 2D manifold (Fig. 5a,c). Typically, these 2D plots contain a large elongated mass corresponding to a combination of background embeddings and the overall effect of dimensional reduction, as well as additional well-defined clusters. These clusters represent common structural moieties in the tomogram including prominently expressed proteins, but also in the case of experimental data, membranes, fiducials and embeddings containing off centered proteins. The clustering workflow of TomoTwin allows users to interactively identify clusters of interest and generate targets for particle picking without requiring any a priori knowledge of the tomogram contents. However, it does require that the abundance of a protein is high enough to form a cluster.

To evaluate the accuracy of clustering-based picking quantitatively, we evaluated the F1 picking score when using clustering-based

picking for each protein in the generalization tomogram (Fig. 5b,e and Extended Data Fig. 7f,g). The clustering-based picking identified each protein with high accuracy across a range of sizes, indicating that this workflow provides an alternative, but effective, method of particle picking. Also notable in the visualized embeddings is the fact that individual protein clusters are globally organized by size, indicating that the model accurately represents similarities in macromolecular shapes as distance in the embedding space (Fig. 5a).

We additionally compared the clustering-based picking against the reference-based approach for the cellular tomograms containing *M. pneumoniae* (Fig. 5c). Visualizing the embeddings of these tomograms, several clusters are visible. One of which, when picked, produces accurate particle locations for 70S ribosomes nearly identical to those produced by the reference-based approach further underlining the robustness of both workflows (Fig. 5d).

Finally, one outstanding question for the study of macromolecules in situ by cryo-ET is how much of the proteome can be feasibly studied by STA. While undoubtably improvements in overall tomogram contrast by denoising[44], missing wedge inpainting[45] or other methods will have a large impact, the clustering workflow of TomoTwin for de novo protein localization represents one method of probing regions of the proteome hitherto uncharted by cryo-ET. To explore the reach of the clustering workflow in situ, we applied it to tomograms containing *Yersinia entomophaga* undergoing controlled cell lysis. After embedding the tomograms, performing the clustering workflow on one tomogram revealed several clusters, two of which corresponded to identifiable biomolecules in the tomograms (Fig. 6a,b). The picked coordinates for each cluster were extracted and used for STA using a spherical reference to assess which biomolecules they contained (Fig. 6c). Refinement of the first cluster (15.5 Å) readily revealed that the cluster contains the 70S bacterial ribosome (Fig. 6d). The second cluster contains a considerably smaller biomolecule. Refinement of these picks by STA with a spherical reference resulted in a low-resolution (18.6 Å) reconstruction of a protein previously undescribed by STA with the approximate shape of a bacterial RNA polymerase (Fig. 6d). RNA polymerase is known to be expressed in high abundance, although at this resolution direct identification of the protein from the refinement is not absolute. Notably, TomoTwin clustered particles of the same protein together regardless of whether they remained contained within the cell or adjacent to it as a result of lysis, demonstrating that the embedding of proteins remains consistent across various surrounding environments.

## Discussion

Despite offering the potential to study proteins in their native, cellular environment, it remains that presently only a select few proteins have been successfully studied by cryo-ET with STA. In part, this is because with increased cellular context, the formation of macromolecular complexes, and poorer contrast caused by thicker specimens, comes the challenge of picking individual proteins for subsequent STA. To assist in this crucial particle picking step, we developed TomoTwin, a robust general picking model for cryo-electron tomograms based on deep metric learning. TomoTwin allows users to identify proteins in tomograms de novo without manually creating training data or retraining the network each time a new protein is to be located.

TomoTwin offers two complementary workflows for picking particles. The reference-based workflow for picking readily observable macromolecules in tomograms, and the clustering workflow that offers a unique opportunity to explore tomogram contents interactively without a priori knowledge. While currently limited in its picking utility by the overall contrast achievable during tomogram reconstruction, the clustering workflow used in conjunction with future advances in tomogram reconstruction presents a potential method of extending the proportion of the proteome that is accessible for STA.

The missing wedge of information in cryo-ETs contributes to the deformation of reconstructed particles particularly along the *xz*

plane[43]. However, in some biological contexts the ability to accurately locate proteins of interest in this view becomes extremely relevant. As opposed to the typical view of a tomogram as a series of *xy* views, TomoTwin picks particles directly in 3D resulting in accurate picking not only in *xy*, but along the *z* axis as well. This allows users to confidently view their picks along their preferred axes of choosing and pick particles that may be difficult to spot when only viewed from the *xy* plane (Extended Data Fig. 4b–f).

Additionally, the need for increased data throughput necessitates the development of algorithms for automated processing in cryo-ET[26,46–49]. TomoTwin can be readily integrated with high throughput tomogram reconstruction and STA workflows and, when combined with unsupervised clustering algorithms[50], TomoTwin paves the way for unsupervised STA analysis on a whole-tomogram level (Extended Data Fig. 9).

While TomoTwin particle picking presents advantages over current workflows in terms of accuracy and usability, it is not without its limitations. Currently, the model is not designed to pick membrane proteins or filaments. Additionally, the appearance of protein clusters in the clustering workflow largely depends on the copy number of the protein in the tomogram. For example, in the *Y. entomophaga* cells, only seven Tc toxin molecules were observed in the tomogram making this protein indiscernible with the clustering workflow, although it is readily and accurately picked with the reference-based workflow. Further, because TomoTwin is trained with a pixel size of 10 Å, the current model is not designed to differentiate between multiple conformations of the same protein at the particle picking level. Finally, although the training data included proteins as small as 30 kDa, during generalization the expected lower limit of accurate picking is approximately 150 kDa on account of TomoTwin picking on downscaled tomograms (roughly 10 Å per pixel). While particles smaller than this size can potentially be located in experimental tomograms, evaluating the accuracy of such picks through STA is, so far, beyond the limitations of the field.

TomoTwin is a robust, open-source tool for particle localization in cryo-electron tomograms that applies the advantages of deep learning while retaining a high degree of usability priming it to assist in a wide range of cryo-ET experiments. The code used to develop and train TomoTwin as well as the general picking model are available at https://github.com/MPI-Dortmund/tomotwin-cryoet.

## Online content

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

# Article

## Methods

### Training data generation

TomoTwin was trained on 123 data classes composed of subvolumes of 120 different proteins, membranes, noise and fiducials from simulated tomograms. To ensure that TomoTwin is trained on the most diverse set of proteins possible, 108 proteins were selected from the PDB with sizes ranging from 30 kDa to 2.7 mDa and the cross-correlation between pairs of 10 Å lowpass filtered maps of each protein was calculated (Extended Data Fig. 2). Any protein with a high similarity (greater than 0.6) to another protein in the training set was marked for replacement. Additionally included were the data from the 2021 SHREC competition[18] including 12 proteins to yield a total of 120 proteins for training. A training/validation split was achieved with 800 subvolumes for each data class in the training set and 200 in the validation set, yielding a total training set size of 98,400 subvolumes and a validation set size of 24,600 subvolumes.

### Tomogram simulation

Tomogram simulation was done using TEM Simulator[51] that calculates the scattering potential of individual proteins and places them in definable positions within the volume. The output of the simulation is a tilt series, which is then reconstructed using IMOD[52]. A configuration file was generated with properties for the electron beam, optics of the microscope, the detector, the tilt geometry and the sample volume. The default detector was adjusted to reflect the MTF curve of a modern Gatan K3 Camera with a quantum efficiency of 0.9. The detector size was set to 1,024 × 1,024 with a pixel size of 5 μm. The magnification was set to 9,800, the spherical aberration and chromatic aberration were adjusted to 2.7 and 2 mm, respectively, to mimic popular modern TEMs. A condenser aperture size of 80 μm was chosen. For each tomogram the defocus value was randomly chosen between −2.5 and −5 μm. A tilting scheme of −60° to +60° with a step size of 2° was used. To simplify and streamline the simulation we wrote a set of open-source programs called 'tem-simulator-scripts' (https://github.com/MPI-Dortmund/tem-simulator-scripts). They contain scripts that require as input the PDB files to be simulated and the number of particles to simulate per PDB. The program then generates reconstructed tomograms as they were used for this study using the following pipeline:

(1) Generation of densely packed random particle positions within the volume where individual particles do not overlap
(2) Generation of an occupancy map: a volume where each voxel is labeled according the protein identity
(3) Generation of fiducial maps
(4) Generation of vesicle maps
(5) Generation of the configuration file for TEM-simulator
(6) Simulation of the tilt series using TEM-simulator
(7) Alignment and reconstruction using IMOD.

However, all steps can also be carried out individually to have full control over all parameters.

Using this procedure, we simulated 11 sets of proteins. The sets contain in total 108 different proteins with each set covering proteins of various sizes. For each set we simulated eight tomograms of size 512 × 512 × 200 voxels with a pixel size of 1.02 nm and varying protein density. For tomogram 1, 2 and 8, 150 particles per protein were generated, for tomograms 3 and 4, 125 particles per protein, for tomograms 5 and 6, 100 particles per protein and for tomogram 7, 75 particles per protein. Tomograms 1–7 were used for training and tomogram 8 for validation. The generated tomograms used in this study with all meta-data are publicly available[53]. These simulated data were used to construct the training and validation sets[54] to evaluate network training, particle localization and model generalizability.

### Convolutional network architecture

To encode volumetric cryo-ET data as embedding vectors in a high-dimensional space, TomoTwin uses a 3D CNN consisting of five convolutional blocks followed by a head network (Extended Data Fig. 1b). Each convolution block consists of two 3D convolutional layers with a kernel size of 3 × 3 × 3. Each convolutional layer is followed by a normalization layer and a leaky rectified linear activation function. In the first convolutional layer of each convolutional block, the number of output channels is twice the input channels and in the second convolutional layer the number of output channels matches the output from the previous layer. Maximum pooling is performed with a kernel size of 2 × 2 × 2 after the first convolutional block and adaptive max pooling to a size of 2 × 2 × 2 is performed after the final convolutional block. As a result, when provided with a 37 × 37 × 37 subvolume with one channel as a normalized, 37 × 37 × 37 × 1 array, the convolutional blocks transform the input to a 2 × 2 × 2 × 1,024 feature vector that is then fed to the head network. In the head network, the feature vector is first flattened channel-wise before being subject to a dropout layer and then passed through a series of fully-connected layers that transform the flattened vector to a 32D feature vector. Finally, this feature vector is L2-normalized to yield an output embedding vector for the subvolume.

### Triplet generation

TomoTwin is trained on triplets of subvolumes consisting of an anchor volume $A$, a positive volume $P$ and a negative volume $N$ (Extended Data Fig. 1c). Each subvolume is assigned to a data class corresponding to the macromolecule contained within and has a size of 37 × 37 × 37 voxels. Triplets are constructed where $A$ and $P$ are sampled from the same data class and $N$ from a different data class. Given a distance function $D$ and an embedding function $f$, the triplet loss is defined as:

$$L(A, P, N) = \max\left(D\left(f(A), f(P)\right) - D\left(f(A), f(N)\right) + a, 0\right)$$

where the hyperparameter $\alpha$ is the margin value. As distance function $D$, we use cosine similarity that is defined as

$$D(Q, P) = \frac{\mathbf{Q} \cdot \mathbf{P}}{\|\mathbf{Q}\| \times \|\mathbf{P}\|}$$

where $\mathbf{Q}$ and $\mathbf{P}$ are arbitrary embedding vectors, • is the dot product and $\|\cdot\|$ the length of the vector. During training, triplets are generated by online semihard triplet mining wherein a batch of subvolumes are embedded and triplets generated automatically with the negative subvolume embedding being selected from those only with a distance to the anchor greater than the positive subvolume embedding but not greater than a margin $\alpha_{\text{miner}}$:

$$D(\mathbf{a}, \mathbf{p}) < D(\mathbf{a}, \mathbf{n}) < d(\mathbf{a}, \mathbf{p}) + \alpha_{\text{miner}}$$

where $\mathbf{a}$, $\mathbf{p}$ and $\mathbf{n}$ are the embedding vectors of the anchor, positive and negative, respectively, and $\alpha_{\text{miner}}$ is the margin of the miner.

### Training of the general picking model

Training of the 3D CNN was performed for 600 epochs using an adaptive moment estimation (ADAM) optimizer[55]. The model from the epoch with the best F1 score on the subvolumes in the validation set was further evaluated in the localization and generalization tasks and used as the general picking model.

### Data augmentation

To prevent overfitting during training and to improve generalization of the model, online data augmentations were applied to each normalized volume before its embedding was calculated including rotation, dropout, translation, and the addition of noise. For the rotation augmentation, subvolumes were rotated by a random angle in the $xy$ plane but not $xz$ or $yz$ to prevent reorientation of the missing wedge. In the dropout augmentation, a random portion between 5 and 20% of the voxels were set to the subvolume mean value. In the translation augmentation, the

subvolume was shifted by 1–2 pixels in each direction. The addition of noise augmentation added Gaussian noise with a randomly chosen standard deviation between 0 and 0.3 to the subvolume.

## Hyperparameter optimization

The training of modern CNNs involves the selection of many hyperparameters, some of these choices affect the architecture while others affect the learning process itself. While some heuristics exist to guide hyperparameter selection, finding a combination of settings that maximize the utility of a machine learning tool by hand quickly becomes intractable. Optuna[56] was applied to explore the hyperparameter search space and identify an optimized set of parameters for training. Models were trained on a subset of the training data for 200 epochs and the F1 score calculated on the validation set after each epoch. Pruning was performed after 50 epochs for training runs with an F1 score lower than the global median. In total, searches were applied for the hyperparameters of learning rate, dropout rate, optimizer, batch size, weight decay, size of the first convolution kernel, number of output layer nodes, online triplet mining strategy (semihard[57], easyhard[58], none), normalization type (group norm[59], batch norm[60]), loss function (TripletLoss[31], SphereFace[61], ArcFace[62]) and loss margin (Extended Data Fig. 10).

Most notably, the type of normalization applied during training was the largest overall affecter of performance with group normalization[59] outperforming the more common batch normalization[60] strategy (Extended Data Fig. 10b). Additionally noted was the increased performance of a standard triplet loss function over the theoretically superior SphereFace[61] and ArcFace[62] loss functions (Extended Data Fig. 10c). These findings underpin the necessity to explore a wide range of hyperparameters during training as heuristics alone are not enough to guide optimal hyperparameter selection for the training of modern CNNs.

## Particle picking workflow with the general model

For each dataset picked with the general model, first all tomograms were embedded. To achieve this, the tomograms were subdivided into a series of overlapping 37 × 37 × 37 subvolumes with a stride of 2 voxels. For the reference-based workflow, a random particle for each protein of interest was selected as reference and embedded to generate a target embedding. The tomogram and target embeddings were provided to TomoTwin Map that calculated the distance matrix between each target embedding and each subvolume embedding from the tomogram and returned this along with a similarity map for each target embedding. This matrix was then provided to TomoTwin Locate that identified areas of high confidence as target locations using a region-growing based maximum detection procedure followed by nonmaxima suppression. The returned candidate positions were then subject to confidence and size thresholding in the TomoTwin graphical user interface to produce final coordinates for each protein of interest.

## Evaluation of simulated data

The performance of particle localization was calculated from three metrics: recall, precision and, the harmonic mean of the two, the F1 score that are defined as:

$$\text{precision} = \frac{\text{true positive}}{\text{true positive} + \text{false positive}}$$

$$\text{recall} = \frac{\text{true positive}}{\text{true positive} + \text{false negative}}$$

$$F1 = 2\frac{\text{precision} \times \text{recall}}{\text{precision} + \text{recall}}$$

Selected particle locations counted as true positives if the intersection over union of the box of the selected particle location and the ground-truth box was greater than 0.6. The intersection over union is defined as the ratio of the intersecting volume of two bounding boxes and the volume of their union.

The particle localization accuracy of the trained model was assessed for each tomogram in the validation set (Extended Data Fig. 3a). To test model generalization, the localization task was performed on a tomogram containing seven proteins not included in the training set for which TomoTwin was therefore naïve (Fig. 2).

## Evaluation of experimental data

Ground-truth particle coordinates are not available for experimental data, which makes the calculation of performance metrics such as precision and recall complicated. One approach to estimate these is to manually count true positives, false positives and false negatives for either the complete tomogram or a reference region.

However, as this is not always possible due to the difficulty of manually picking some proteins, another approach to measure the picking precision without introducing reference bias is by extracting subvolumes at the picked coordinates of each protein, projecting the 3D subvolumes to 2D using SPHIRE[63] and performing reference-free 2D classification[46]. 2D classification on few particles can lead to extraneous results because only the most common poses of a protein will have enough particles to be clustered effectively, possibly resulting in many true-positive particles classified incorrectly as false positives. Nevertheless, the percentage of particles in good 2D classes was measured as an indicator of the precision.

## Clustering

For clustering analysis, a random sample of 400,000 embeddings from the high-dimensional tomogram embeddings were fit to a uniform 2D manifold with uniform manifold approximation (UMAP) with GPU-acceleration provided by the RAPIDS package[64]. The UMAP model was used as the basis to transform the entire tomogram embeddings and the results plotted (Fig. 5a,c). Clusters were identified by eye and selected by drawing a closed shape containing the desired points. The enclosed points were then traced back to their original high-dimensional embeddings and the average embedding of them was calculated. This average embedding was then used as a target embedding for classification, localization and picking in the same manner as for the reference-based workflow.

## Preparation of experimental samples

The components of the mixture were either thawed from long-term storage at −80 °C or freshly prepared. *P. luminescens* holotoxin was expressed, purified and the holotoxin formed as described previously[65] and used at a stock concentration of 0.49 mg ml⁻¹. RhsA from *P. protegens* was expressed and purified as described previously[35] and used at 4 mg ml⁻¹ concentration. Liposomes were prepared by extrusion. 4 mg ml⁻¹ of each POPC (1-palmitoyl-2-oleoyl-glycero-3-phosphocholine, Avanti Polar Lipids) and DOPS (1,2-dioleoyl-sn-glycero-3-phospho-L-serine, Avanti Polar Lipids) were mixed in buffer (50 mM Tris, pH 8, 150 NaCl, 0.05% Tween20) and after brief sonication (1 min in water bath) and three cycles of freeze–thawing (−196 °C and 50 °C), the liposome solution was passed 11 times through a polycarbonate membrane with a 400 nm pore size in a mini extruder (Avanti Polar Lipids). Total lipid concentration was diluted with buffer to 0.16 mg ml⁻¹. The freeze-dried content of one vial tobacco mosaic virus (DSMZ GmbH Braunschweig, Germany, PC-0107) was solved in 1 ml of buffer and diluted 500 times as working solution. The apoferritin plasmid was a kind gift from C. Savva (Midlands Regional Cryo-Electron Microscopy Facility). Expression and purification of apoferritin was optimized based on the protocol described earlier[34] and final concentration of frozen stock was 3 mg ml⁻¹.

Different ratios of the mixture were prepared and then examined after vitrification using cryo-EM. For cryo-ET, a mixture ratio of

1:2:2:20:10 (tobacco mosaic virus:apoferritin:liposomes:Tc toxin:RhsA) was chosen.

## Grid preparation

Grids were prepared using a Vitrobot Mark IV (Thermo Fisher Scientific) at 4 °C and 100% humidity. Then 4 µl of the freshly prepared mixture were applied to glow-discharged (Quorum GloQube) R1.2/1.4 Cu 200 (Quantifoil) grids. After blotting (3.5 s at blot force −1, no drain time) the specimen was vitrified in liquid ethane.

## Cryo-ET

Grids of different mixing ratios were screened using a Talos Arctica electron microscope (Thermo Fisher Scientific) equipped with a X-FEG and Falcon 3 camera. Small datasets of 100–200 images were collected using the software EPU (Thermo Fisher Scientific). The best specimen was transferred to a Titan Krios G3 electron microscope equipped with X-FEG. Images were recorded on a K3 camera (Gatan) operated in counting mode at a nominal magnification of 63,000, resulting in a pixel size of 1.484 Å per pixel. A Bioquantum post-column energy (Gatan) was used for zero loss imaging with a slit width of 20 eV.

Tilt series were acquired using SerialEM[66] with the Plugin PACE-tomo[67] and with a dose symmetric tilt scheme[68] from −60° to 60° with a step size of 3°. Each movie was collected as an exposure of 0.2 s subdivided into ten frames. Frames were then exported to Warp v.1.0.9 (ref. 26) for motion correction, contrast transfer function estimation and generation of tilt series. Tilt series were aligned with patch tracking and tomograms reconstructed by weighted back projection in IMOD[52] with a pixel size of 5.936. Tomograms were scaled by Fourier shrinking to 10 Å per pixel for embedding with TomoTwin.

Raw frames of *M. pneumoniae* cells were downloaded from EMPIAR (EMPIAR 10499). Motion correction and contrast transfer function estimation were performed in Warp v.1.0.9, which was then used to generate tilt series. These tilt series were aligned with patch tracking and tomograms reconstructed by weighted back projection in IMOD with a pixel size of 6.802 Å per pixel. Tomograms were then scaled by Fourier shrinking to 13.6 Å per pixel for embedding with TomoTwin.

## Evaluation of experimental data

For tomograms from samples prepared in-house, coordinates of particles identified with TomoTwin were scaled to a pixel size of 5.936 to match the originally reconstructed tomograms. The tomograms were imported and these coordinates were used to extract subtomograms in Relion v.3.0 (ref. 46). For reference-free analysis, 3D subtomograms were projected to 2D with SPHIRE[63] and then used for 2D classification.

Tomograms containing *M. pneumoniae* were attained from EMPIAR (EMPIAR 10499), coordinates of particles identified with TomoTwin were scaled to a pixel size of 6.802 Å per pixel to match the originally reconstructed tomograms. The tomograms were imported and coordinates were imported and used to reconstruct pseudo-subtomograms in Relion 4.0 (ref. 38). A reference was created from a 70S ribosome (EMD-11650) by lowpass filtering to 30 Å and then scaling the pixel size to 6.802 Å per pixel. This reference was used for 3D classification with the pseudo-subtomograms in Relion v.4.0.

The tilt series and alignment files for the tomogram containing *Chlamydomonas reinhardtii* cells was attained from EMPIAR (EMPIAR 10694) and used to reconstruct the tomogram at a pixel size of 13.68 Å per pixel for picking and 6.84 Å per pixel for STA. Picking of RuBisCO was performed using the TomoTwin reference-based workflow and the resulting coordinates scaled to the pixel size to be used for extraction and, subsequently, STA in Relion v.3.0 (ref. 46). Extracted subtomograms were used for initial 3D refinement using a reference generated by lowpass filtering a known model of RuBisCO (PDB 1BXN) to 60 Å. This map was used to fit the model of RuBisCO.

Reconstructed tomograms were binned to 9.288 Å per pixel and picked with the TomoTwin clustering workflow. Picked coordinates for each cluster were rescaled to 4.644 Å per pixel for extraction. Tomograms were imported and these coordinates used to extract subtomograms in Relion v.3.0 (ref. 46). An initial 3D refinement was performed with a spherical reference to eliminate the possibility of reference bias, and the resulting map used for 3D classification without alignment. Finally, 3D refinement was repeated using one of the 3D classes as reference to achieve a coherent 3D refined map that was then used to fit the PDB models of candidate proteins.

## Hardware

Two computational setups were used for calculations, a distributed computing system and a local workstation. The distributed computing system consisted of the Max Planck Gesellschaft Supercomputer 'Raven' using up to 30 Nvidia A100 graphical processing units (GPUs), where each GPU has 40 GB memory. Each process had 18 cores of Intel Xeon IceLake-SP 8360Y processors and 128 GB system memory available. The local workstation consisted of a local unit equipped with a Nvidia Titan V (12 GB memory) GPU and an Intel i9-7920X CPU with 64 GB system memory.

Hyperparameter optimization was done in parallel for 7 d once the distributed computing setup and embeddings were calculated on this, also using two GPUs. In all cases, a box size of 37 and stride of two were used for embedding.

The in-house workstation was used for miscellaneous tasks and for calculating timings using two GPUs.

## Timings

The calculation of the embeddings is the only function of TomoTwin requiring notable processing time. To measure this, we embedded our largest experimental tomogram (608 × 855 × 148 after Fourier shrinking) on a local workstation and a distributed computing system. Using two GPUs, tomogram embedding took 80 min for the local setup and 30 min for the distributed setup, corresponding to the total time to pick all proteins of interest per tomogram on each setup.

## Statistics and reproducibility

Embeddings produced by the general model are deterministic allowing users to reproduce picking results at will. The only package used that is not deterministic is the Nvidia RAPIDS UMAP that is used for visualization in the clustering workflow. However, as picking and thresholding calculations are always performed on the original embeddings, the overall picking results remain consistent.

## Reporting summary

Further information on research design is available in the Nature Portfolio Reporting Summary linked to this article.

## Data availability

All simulated tomograms and subvolumes used to train and evaluate the performance of TomoTwin are available at https://doi.org/10.5281/zenodo.6637357 (tomograms) and https://doi.org/10.5281/zenodo.6637456 (subvolumes).

## Code availability

The TEM-Simulator-Scripts package used for automated tilt-series simulation and reconstruction is available at https://github.com/MPI-Dortmund/tem-simulator-scripts. TomoTwin is available under an open-source license at https://github.com/MPI-Dortmund/tomotwin-cryoet. A code demonstration for TomoTwin complete with system requirements, installation instructions, demonstration scripts, data and usage instructions is available at https://doi.org/10.5281/zenodo.7186070.

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

## Acknowledgements

We thank P. Günther for providing RhsA and liposomes, P. Njenga Ng'Ang'A for providing TcdA1, Z. Wang for collecting and reconstructing tomograms containing YenTc, K. Vogel-Bachmayr for purifying apoferritin, C. Savva for providing the apoferritin plasmid and A. Prajica for support in figure and logo creation. The work was supported by the Max Planck Society (to S.R.).

## Author contributions

Conceptualization was done by T.W. and S.R. Software implementation was done by T.W., G.R. and M.S. Software testing was done by G.R., T.W. and M.S. Data acquisition and sample preparation were done by G.R., O.S. and D.P. Formal analysis was done by G.R. and T.W. Supervision was carried out by S.R. The original draft was written by G.R. and T.W. Review and editing of the paper was done by G.R., T.W. and S.R. Funding was acquired by S.R.

## Funding

## Competing interests

The authors declare no competing interests.

## Additional information

**Extended data** is available for this paper at https://doi.org/10.1038/s41592-023-01878-z.

**Correspondence and requests for materials** should be addressed to Stefan Raunser.

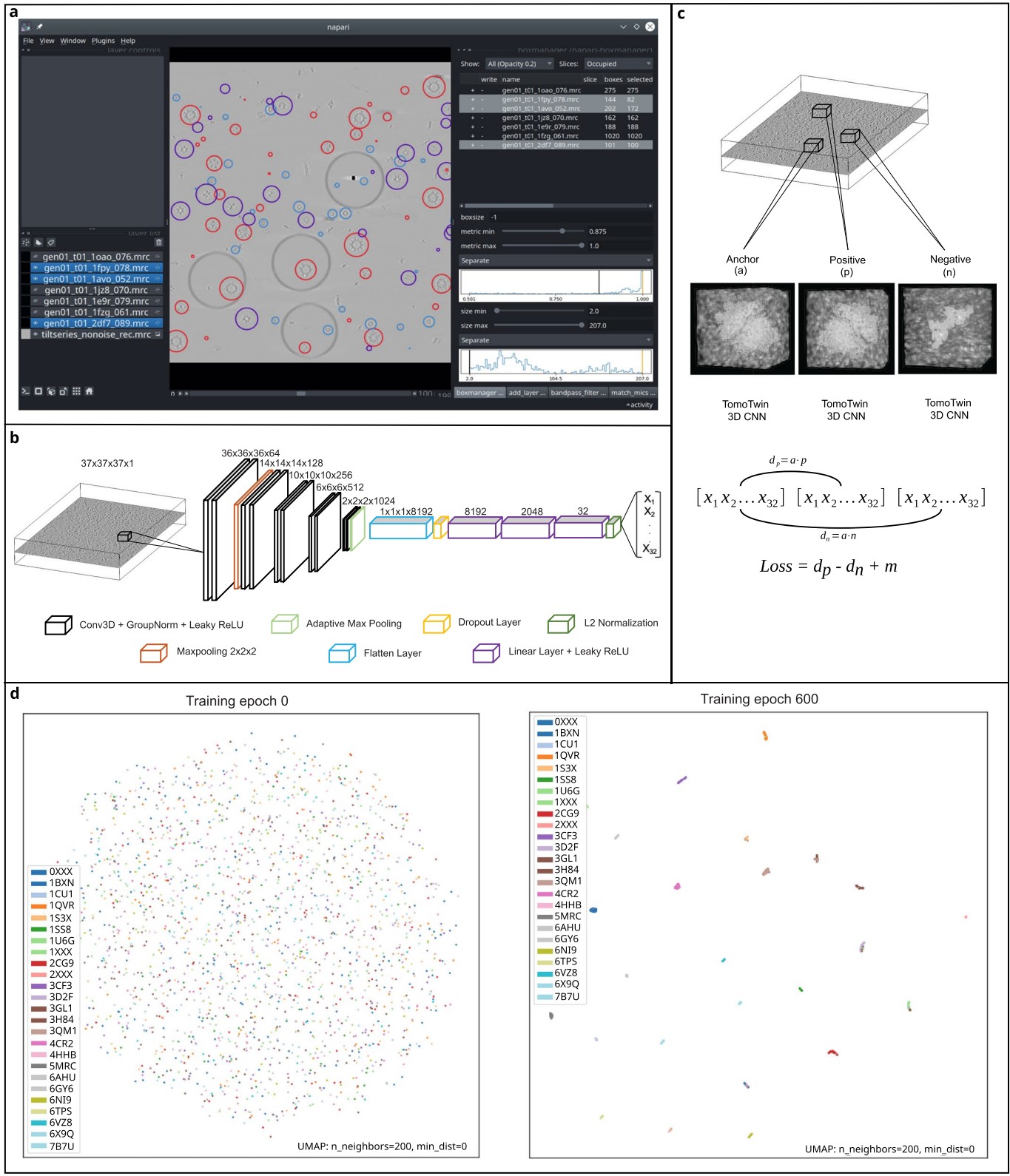

**Extended Data Fig. 1 | TomoTwin convolutional architecture and metric learning strategy for particle picking. a**, TomoTwin user interface in Napari for visualizing protein picks. Picks for 3 proteins are shown as spheres. Lefthand panel allows users to adjust visualization settings for the. Righthand panel to filter picks for each cluster according to similarity threshold, minimum and maximum size, and adjust the box size for viewing. **b**. Architecture of 3D convolutional network utilized by TomoTwin to embed tomogram subvolumes for deep metric learning. **c**. Overview of the deep metric learning training scheme wherein data triplets are constructed of anchor, positive, and negative subvolumes. The subvolumes are each convolved by the CNN and the embeddings are used to calculate the distance metrics in the triplet loss function. **d**. UMAP of protein subvolume embeddings colored according to protein PDB ID from TomoTwin 3D CNN in first training epoch and best model after 600 training epochs.

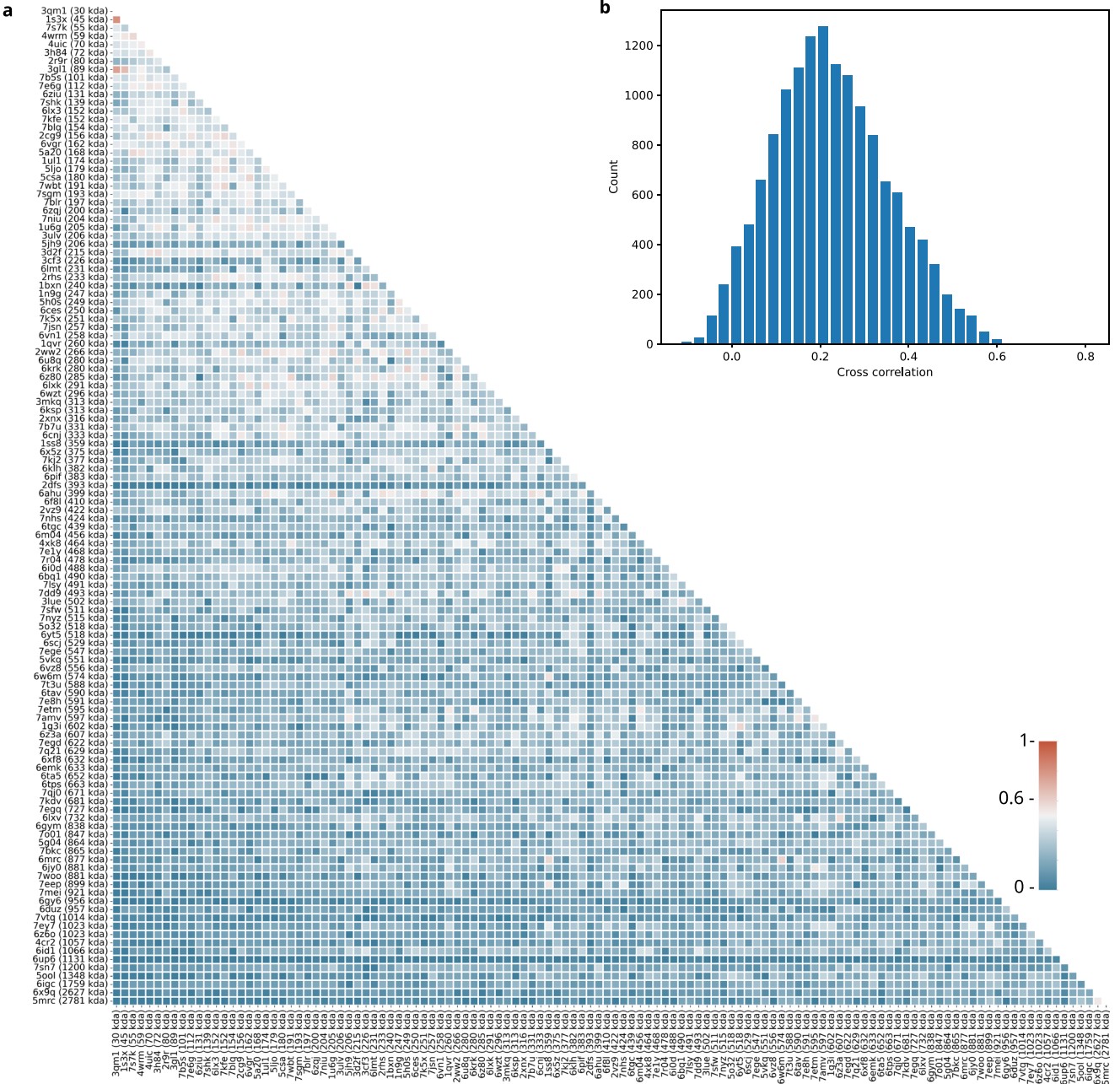

**Extended Data Fig. 2 | Characterization of the training dataset. a**, Pairwise cross-correlation matrix for all 120 proteins sorted by size. Cross-correlations were calculated by converting the individual PDBs to density maps with a pixel size of 1 nm, aligning them pairwise with EMAN2 and calculating the cross-correlation of the aligned pairs. To maximize the value for training, we selected proteins so that all pairs except 3 have a cross-correlation value below 0.6. The three pairs with higher correlation are from the SHREC dataset and were not simulated by us. Higher correlation values are more likely for smaller proteins. **b**, Histogram of the pairwise cross correlation values. The mean cross correlation value is 0.22 with a standard deviation of 0.13.

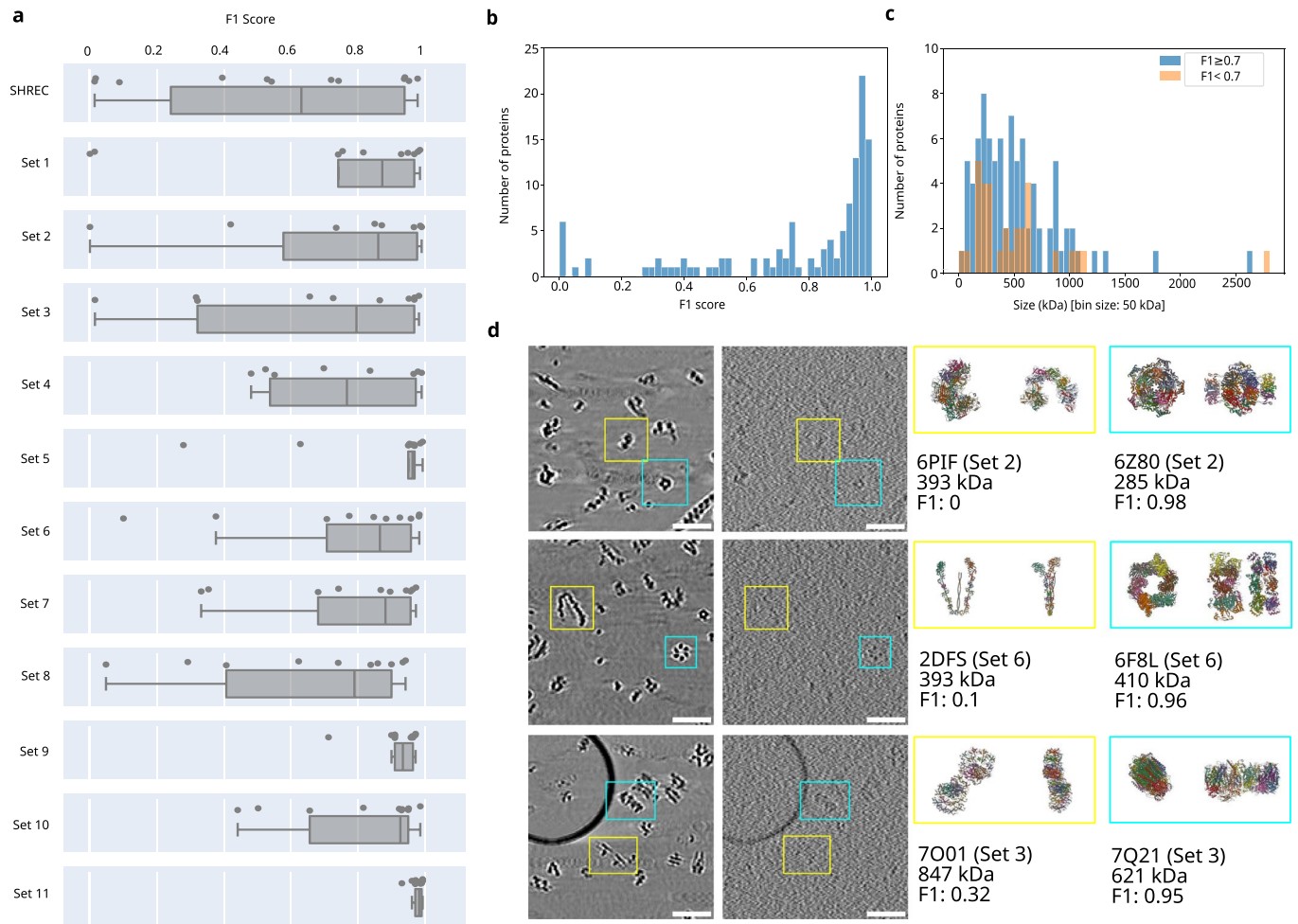

**Extended Data Fig. 3 | TomoTwin identifies proteins with high accuracy by using single particle subvolumes as reference. a**, F1 scores of TomoTwin on the validation tomograms. Each box extends from the 1st quartile (Q1) to the 3rd quartile (Q3). The median is marked by a line inside the box. Whisker lines correspond to box edges +/− 1.5 times interquartile range. The number of proteins n in each set is 10, except for SHREC (n = 12) and Set 2 and 4 (n = 8). The median F1 score of the individual sets is most often above 0.8 and not lower than 0.76. **b**, The overall distribution of F1 scores for the complete training set.

The distribution has a median of 0.92, but also a tail of low F1 scores can be seen. **c**, Size distribution of particles that show good F1 scores (F1 > = 0.7) and those with rather low F1 scores (F1 < 0.7). **d**, Examples of proteins of similar size with low (yellow) and high (cyan) F1 score. On the left side the individual particles are depicted in a noisy and noise-free reconstruction, respectively. On the right side, the respective structures, PDB IDs, sizes, and F1 scores are shown. The proteins which were not identified properly by TomoTwin have a lower contrast than the other proteins. Scale bars 100 nm.

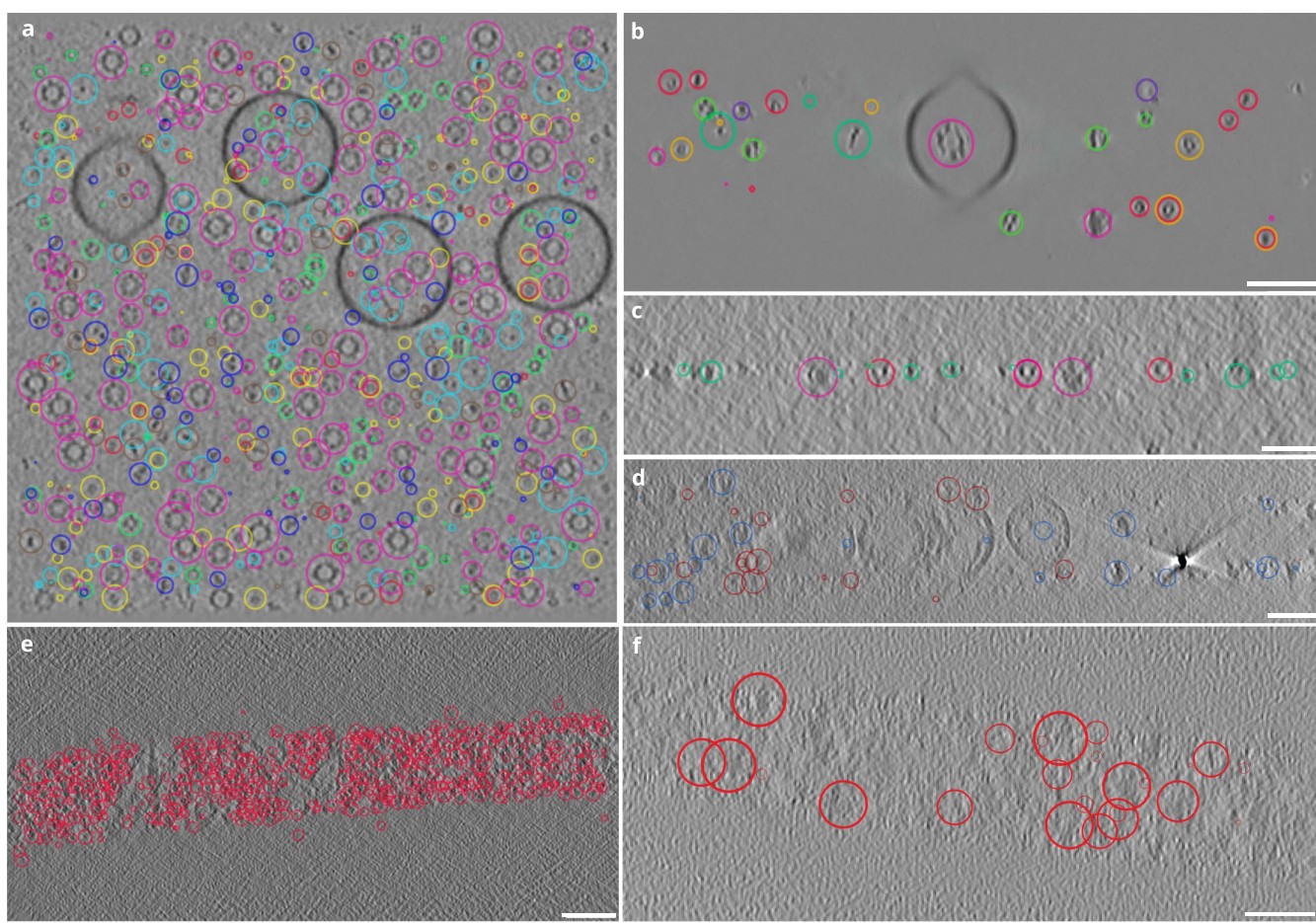

**Extended Data Fig. 4 | TomoTwin locates particles accurately in densely packed tomograms and with Z-centering. a. Picking in** simulated dense generalization tomogram for PDB IDs: 1AVO (red), 1E9R (yellow), 1FPY (green), 1FZG (light blue), 1JZ8 (dark blue), 1OAO (brown), and 2DF7 (magenta). **b.** XZ view of picking simulated generalization tomogram (color scheme: PDB ID: 1AVO (red); PDB ID: 1E9R (orange); PDB ID: 1FPY (green); PDB ID: 1FZG (teal); PDB ID: 1JZ8 (blue); PDB ID: 1OAO (purple); PDB ID: 2DF7 (magenta)). Scale bar 50 nm.

**c**. XZ view of picking tomogram containing a pyrenoid inside a *C. reinhardtii* cell with RuBisCO picked. Scale bar 75 nm. **d**. XZ view of picking tomogram containing a *M. pneuomoniae* cell with ribosomes picked. Scale bar 50 nm. **e**. XZ view of picking tomogram containing a mixture of apoferritin (red), TcdA1 (magenta), and RhsA (teal). Scale bar 50 nm. **f.** XZ view of picking tomogram containing a *Y. entomophaga* cell with ribosomes (blue) and RNA polymerases (red) picked. Scale bar 75 nm.

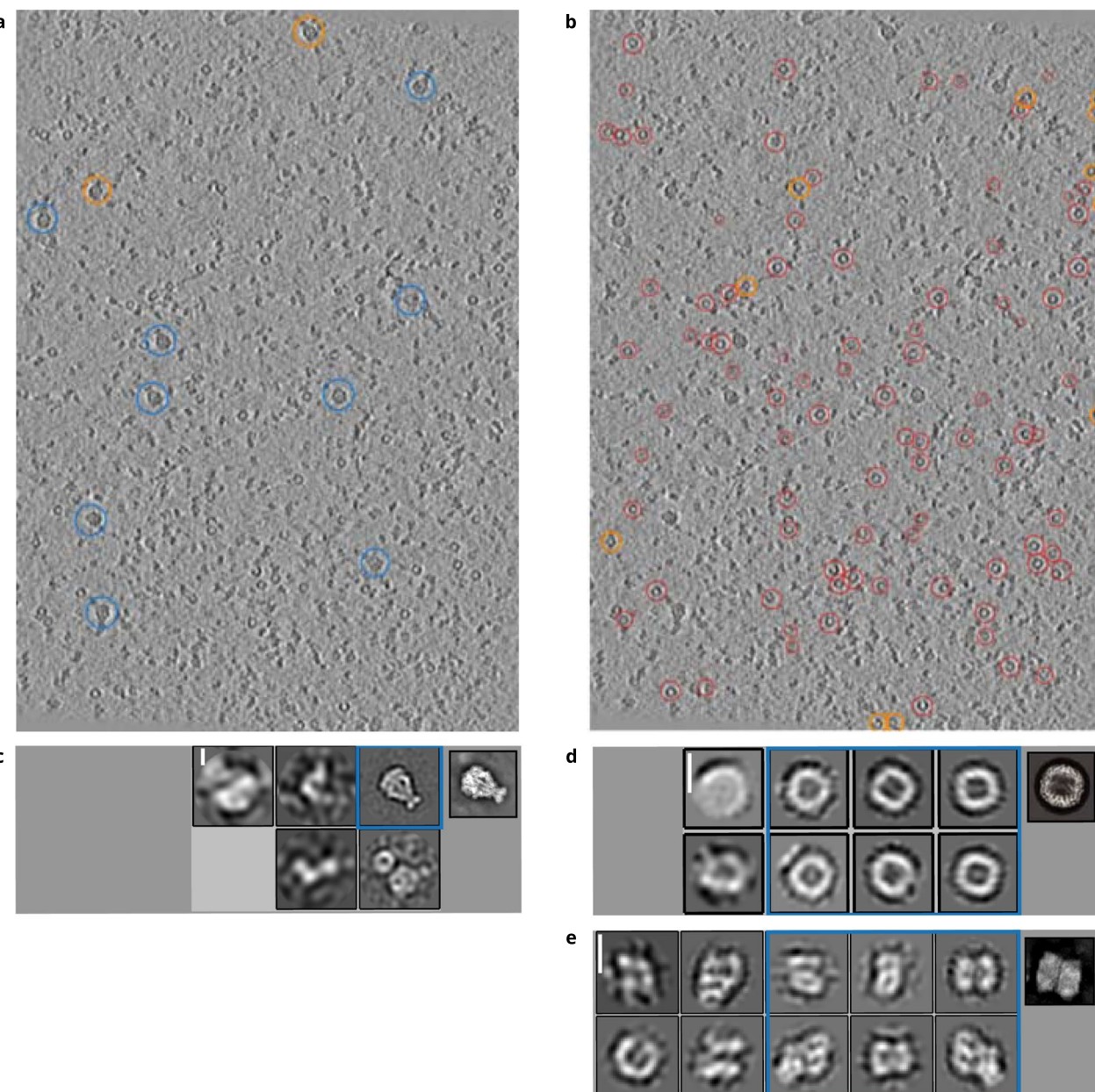

**Extended Data Fig. 5 | Picking evaluation of in-vitro tomogram.** The accuracy was evaluated on one full tomogram for TcdA1 and apoferritin by manual counting of picked and missed particles. For TcdA1 (**a**.) 9 particles were picked (blue) and two particles were missed (orange). For apoferritin (**b**.) 126 particles were picked (red) and 11 particles missed (orange). The results give an estimated recall of 0.81 and 0.91 for TcdA1 and apoferritin respectively. There were no false positives in either case (precision 1.0). A manual evaluation for RhsA was not feasible by eye. Example 2D classes from previous studies by single particle analysis of **c**. apoferritin[34], **d**. RhsA[35], and **e**. TcdA1[36]; 2D class averages of TomoTwin picked subvolumes after projection to 2D. Classes outlined in blue were judged to be positive classes by expert inspection, indicating that they contain particles of the appropriate protein. Scale bar: 5 nm.

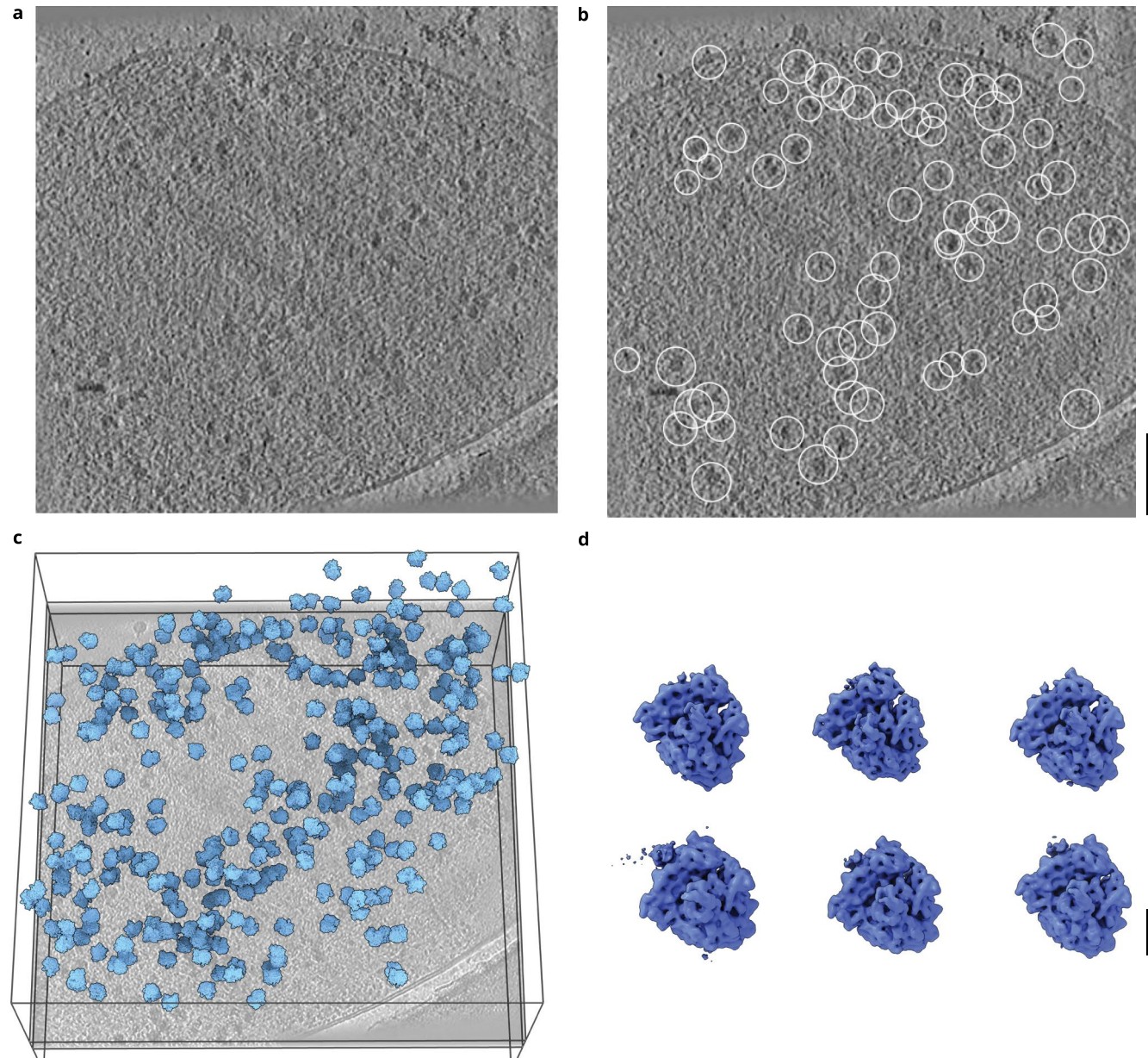

**Extended Data Fig. 6 | TomoTwin locates proteins in a cellular environment.** **a**, Representative slice view of a tomogram containing *Mycoplasma pneumoniae*. b. Slice view highlighting positions of picked 70S ribosomes localized in 3D with TomoTwin. Scale bar 100 nm. **c**, 3D representation of ribosome positioning within the tomogram, a represented slice is superimposed with 3D classes of ribosomes arranged according to their corresponding coordinates and orientation. **d**, 3D classes from 18,246 particles. Scale bar 10 nm.

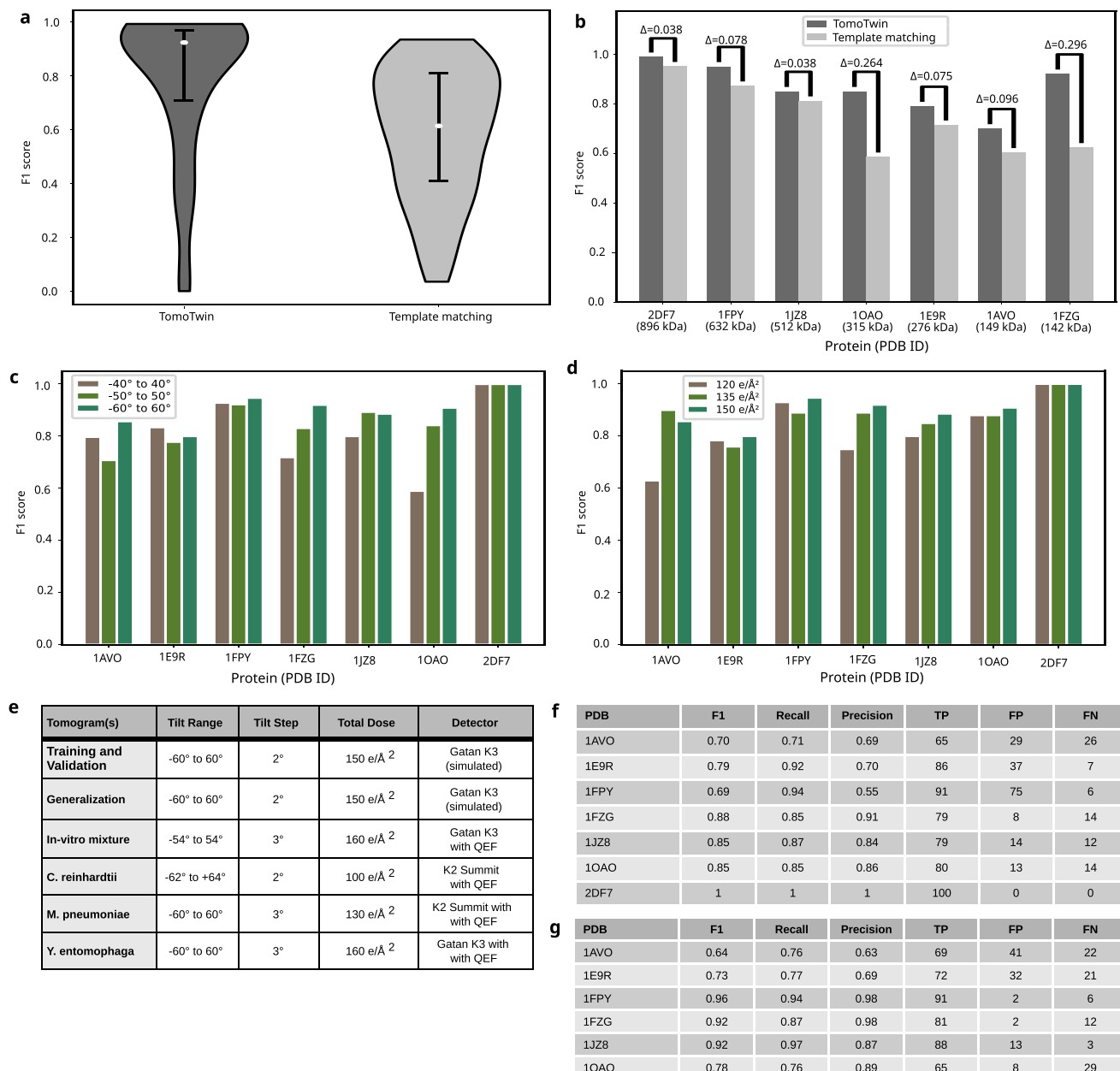

**Extended Data Fig. 7 | TomoTwin outperforms template matching and adapts to picking across a variety of experimental parameters. a**, F1 score benchmark for TomoTwin and template matching of n = 120 proteins on the respective validation tomograms. For TomoTwin the results of the reference mode with a single subvolume as reference are shown. TomoTwin similarity and size threshold were optimized for each protein. For template matching the EMAN2 implementation with a density as reference generated by the respective protein PDB was used. The median F1 score is represented by a white dot and the 25th and 75th percentile by the indicated range. Width of plots indicates the density of F1 scores across the dataset. **b**, Detailed comparison for the generalization tomogram. TomoTwin and template matching were applied analogously to (**a**). In contrast, the best results of either the reference mode or the clustering mode are shown for TomoTwin. **c**, F1 picking score for each protein in the simulated generalization tomogram reconstructed from tilt series with varying tilt ranges. **d**, F1 picking score for each protein in the simulated generalization tomogram with varying total electron doses. **e**, Tilt range, tilt step, total dose, and detector used for each dataset picked with TomoTwin. Detailed picking statistics including F1 score, recall, precision, true positives (TP), false positives (FP) and false negatives (FN) for the generalization tomogram picked by **f**. reference workflow and **g**. clustering workflow.

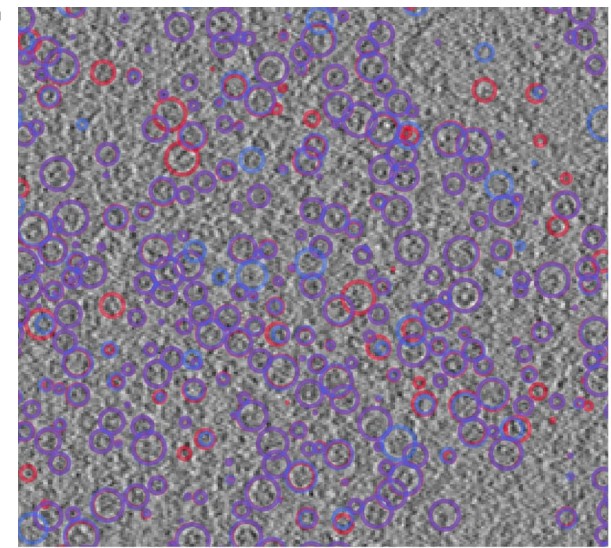

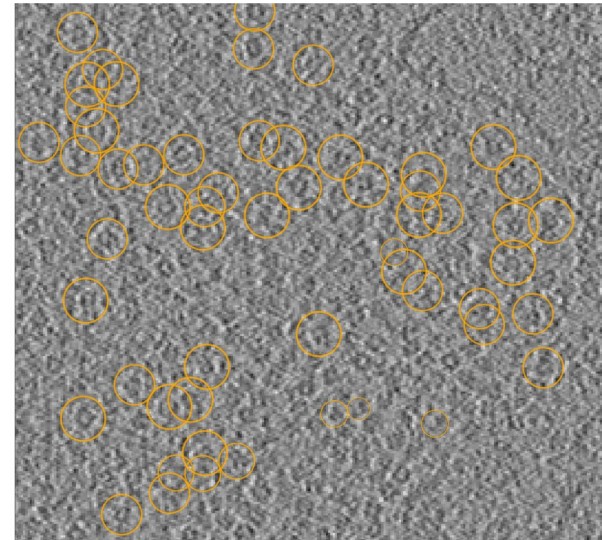

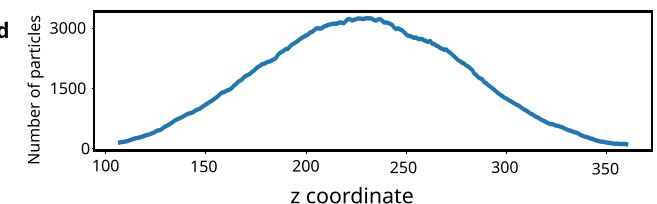

| Method | Total | Unique | Missed |
|--------|-------|--------|--------|
| Reference | 257 | 21 | 57 |
| Clustering | 249 | 13 | |

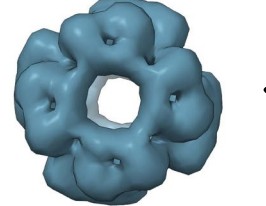 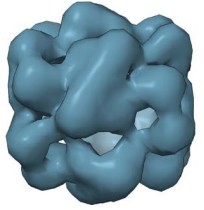 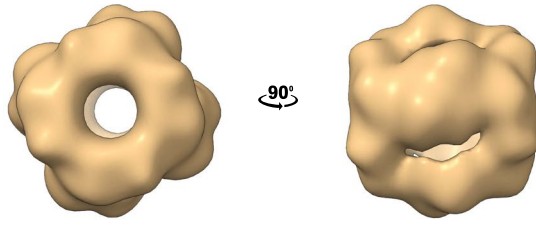

**Extended Data Fig. 8 | Additional evaluation statistics for the RuBisCO tomogram using the reference- and clustering-based workflows. a**, Color coded picks for a selected sub-region where purple picks were picked by both workflows, red by the reference workflow only and blue by the clustering workflow only. For both methods the threshold was set to get a visually optimal result. The threshold was 0.91 for both workflows. **b**, Missed particles by both workflows. **c**, Summary statistics for both workflows on the selected sub-region. Based on this we estimate that 80% of the particles were picked. **d**, The number of picked particles follows the particle density along the z-direction of the tomogram. Scale bars 50 nm. **e**. Comparison between *in situ* subtomogram averages of RuBisCO from EMDB-3694[42] at 16.5 Å (bottom) and the same tomogram picked with TomoTwin at 13.7 Å (top).

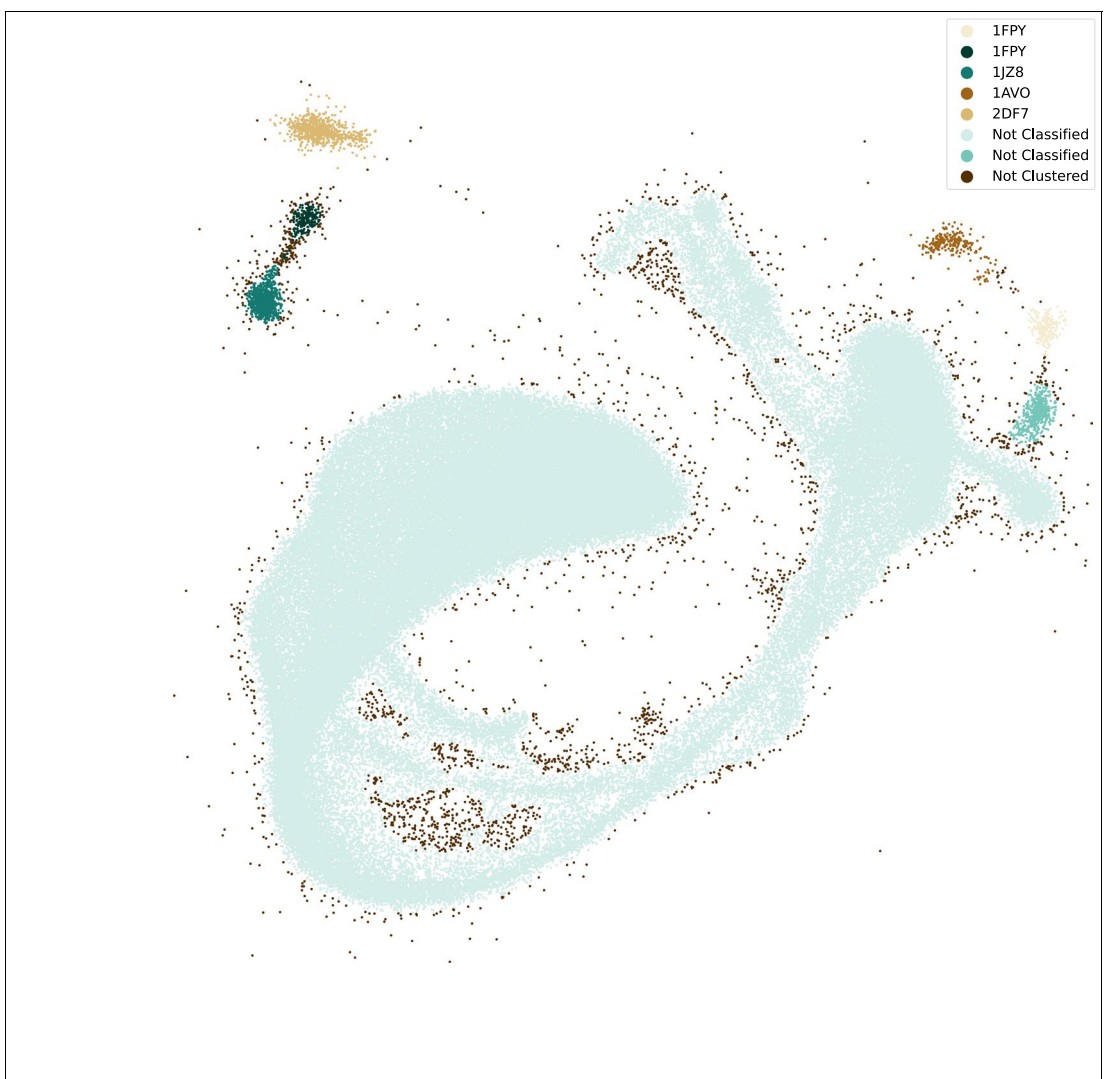

**Extended Data Fig. 9 | Automated identification of clusters of interest using HDBSCAN.** A subset of the approximated manifold of Fig. 5a was used to run density-based clustering which located 5 out of 7 proteins of interest in an unsupervised manner. R implementation of HDBSCAN was run with a min_samples of 50 and a minimum cluster size of 50.

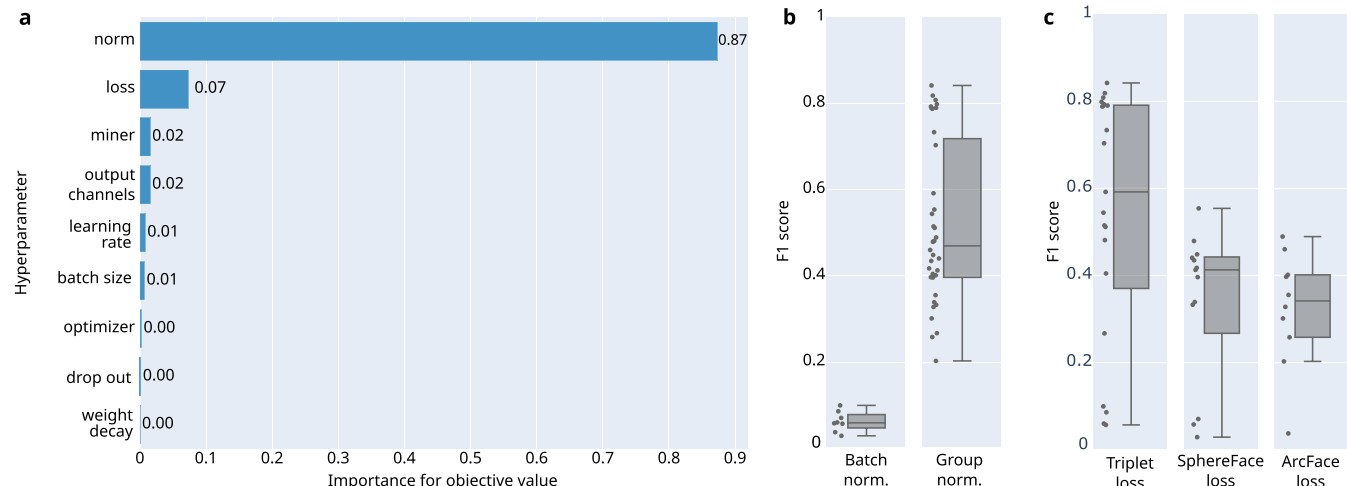

**Extended Data Fig. 10 | Hyperparameter optimization of TomoTwin.**
**a**, Hyperparameter importance estimated by Optuna[56] after 180 trials with different configurations. **b**, F1 scores for trials using either the batch normalization (n = 8) or group normalization layers (n = 36) in convolutional neural network. Group normalization performed better than batch normalization in all cases. **c**, F1 score for trials using either Triplet-, SphereFace-, or ArcFace-Loss (n = 19, 13, or 10 respectively). The boxes in b and c extend from the 1st quartile (Q1) to the 3rd quartile (Q3). The median is marked by a line inside the box. Whisker lines correspond to box edges +/− 1.5 times interquartile range. Points represent the individual trials.

# Reporting Summary

## Statistics

For all statistical analyses, confirm that the following items are present in the figure legend, table legend, main text, or Methods section.

| n/a | Confirmed | |
|---|---|---|
| ☐ | ☒ | The exact sample size (*n*) for each experimental group/condition, given as a discrete number and unit of measurement |
| ☒ | ☐ | A statement on whether measurements were taken from distinct samples or whether the same sample was measured repeatedly |
| ☒ | ☐ | The statistical test(s) used AND whether they are one- or two-sided<br>*Only common tests should be described solely by name; describe more complex techniques in the Methods section.* |
| ☒ | ☐ | A description of all covariates tested |
| ☒ | ☐ | A description of any assumptions or corrections, such as tests of normality and adjustment for multiple comparisons |
| ☐ | ☒ | A full description of the statistical parameters including central tendency (e.g. means) or other basic estimates (e.g. regression coefficient) AND variation (e.g. standard deviation) or associated estimates of uncertainty (e.g. confidence intervals) |
| ☒ | ☐ | For null hypothesis testing, the test statistic (e.g. *F*, *t*, *r*) with confidence intervals, effect sizes, degrees of freedom and *P* value noted<br>*Give P values as exact values whenever suitable.* |
| ☒ | ☐ | For Bayesian analysis, information on the choice of priors and Markov chain Monte Carlo settings |
| ☒ | ☐ | For hierarchical and complex designs, identification of the appropriate level for tests and full reporting of outcomes |
| ☒ | ☐ | Estimates of effect sizes (e.g. Cohen's *d*, Pearson's *r*), indicating how they were calculated |

*Our web collection on statistics for biologists contains articles on many of the points above.*

## Software and code

Policy information about availability of computer code

| | |
|---|---|
| Data collection | - TEM-Simulator: http://tem-simulator.sourceforge.net/ (Version 1.3)<br>- TEM-Simulator-Scripts: https://github.com/MPI-Dortmund/tem-simulator-scripts |
| Data analysis | - TomoTwin: https://github.com/MPI-Dortmund/tomotwin-cryoet (0.2.2)<br>- SPHIRE: https://sphire.mpg.de/ (Version 1.4)<br>- EMAN2 (Version 2.3)<br>- RELION: https://relion.readthedocs.io/ (Version 3.1) |

For manuscripts utilizing custom algorithms or software that are central to the research but not yet described in published literature, software must be made available to editors and reviewers. We strongly encourage code deposition in a community repository (e.g. GitHub). See the Nature Portfolio guidelines for submitting code & software for further information.

## Data

Policy information about availability of data

All manuscripts must include a data availability statement. This statement should provide the following information, where applicable:
- Accession codes, unique identifiers, or web links for publicly available datasets
- A description of any restrictions on data availability
- For clinical datasets or third party data, please ensure that the statement adheres to our policy

All simulated tomograms used in this study are available here: https://doi.org/10.5281/zenodo.6637357. The extracted subvolumes used to train and evaluate the performance of TomoTwin are available at: https://doi.org/10.5281/zenodo.6637456.

## Human research participants

Policy information about studies involving human research participants and Sex and Gender in Research.

| | |
|---|---|
| Reporting on sex and gender | n/a |
| Population characteristics | n/a |
| Recruitment | n/a |
| Ethics oversight | n/a |

Note that full information on the approval of the study protocol must also be provided in the manuscript.

# Field-specific reporting

Please select the one below that is the best fit for your research. If you are not sure, read the appropriate sections before making your selection.

☒ Life sciences  ☐ Behavioural & social sciences  ☐ Ecological, evolutionary & environmental sciences

For a reference copy of the document with all sections, see nature.com/documents/nr-reporting-summary-flat.pdf

# Life sciences study design

All studies must disclose on these points even when the disclosure is negative.

| | |
|---|---|
| Sample size | In this study, no conclusions based on statistical tests were drawn. Only descriptive statistics were used. In terms of the training data size, we increased it until the network performance saturated. |
| Data exclusions | No data was excluded |
| Replication | The general model that is used for picking works deterministic. As long the same model is used, the results are replicable. Therefore all findings in this study are replicable. |
| Randomization | Not relevant, as no samples that belong to certain experimental groups exist. |
| Blinding | There was no group allocation, therefore not blinding was necessary or possible. |

# Reporting for specific materials, systems and methods

We require information from authors about some types of materials, experimental systems and methods used in many studies. Here, indicate whether each material, system or method listed is relevant to your study. If you are not sure if a list item applies to your research, read the appropriate section before selecting a response.

## Materials & experimental systems

| n/a | Involved in the study |
|-----|----------------------|
| ☒ | Antibodies |
| ☒ | Eukaryotic cell lines |
| ☒ | Palaeontology and archaeology |
| ☒ | Animals and other organisms |
| ☒ | Clinical data |
| ☒ | Dual use research of concern |

## Methods

| n/a | Involved in the study |
|-----|----------------------|
| ☒ | ChIP-seq |
| ☒ | Flow cytometry |
| ☒ | MRI-based neuroimaging |

