## [Peer Review File · Nature Methods]

Peer Review Information

Manuscript Title: TomoTwin: Generalized 3D Localization of Macromolecules in Cryo-electron Tomograms with Structural Data Mining

Corresponding author name(s): Stefan Raunser

Editorial Notes: n/a

Reviewer Comments & Decisions:

Decision Letter, first revision:

Dear Stefan,

Your Article, "TomoTwin: Generalized 3D Localization of Macromolecules in Cryo-electron Tomograms with Structural Data Mining", has now been seen by 3 reviewers. As you will see from their comments below, although the reviewers find your work of considerable potential interest, they have raised a number of concerns.

We are interested in the possibility of publishing your paper in Nature Methods, but would like to consider your response to these concerns before we reach a final decision on publication. We therefore invite you to revise your manuscript to address these concerns.

* include a point-by-point response to the reviewers and to any editorial suggestions

* please underline/highlight any additions to the text or areas with other significant changes to facilitate review of the revised manuscript

- * address the points listed described below to conform to our open science requirements
- * ensure it complies with our general format requirements as set out in our guide to authors at www.nature.com/naturemethods
- * resubmit all the necessary files electronically by using the link below to access your home page

[Redacted] This URL links to your confidential home page and associated information about manuscripts you may have submitted, or that you are reviewing for us. If you wish to forward this email to co-authors, please delete the link to your homepage.

We hope to receive your revised paper within 6 weeks. If you cannot send it within this time, please let us know. In this event, we will still be happy to reconsider your paper at a later date so long as nothing similar has been accepted for publication at Nature Methods or published elsewhere.

OPEN SCIENCE REQUIREMENTS

REPORTING SUMMARY AND EDITORIAL POLICY CHECKLISTS

Please note that these forms are dynamic 'smart pdfs' and must therefore be downloaded and completed in Adobe Reader. We will then flatten them for ease of use by the reviewers. If you would

like to reference the guidance text as you complete the template, please access these flattened versions at <http://www.nature.com/authors/policies/availability.html>.

DATA AVAILABILITY

We strongly encourage you to deposit all new data associated with the paper in a persistent repository where they can be freely and enduringly accessed. We recommend submitting the data to discipline-specific and community-recognized repositories; a list of repositories is provided here:

<http://www.nature.com/sdata/policies/repositories>

All novel DNA and RNA sequencing data, protein sequences, genetic polymorphisms, linked genotype and phenotype data, gene expression data, macromolecular structures, and proteomics data must be deposited in a publicly accessible database, and accession codes and associated hyperlinks must be provided in the “Data Availability” section.

Please include a “Data availability” subsection in the Online Methods. This section should inform readers about the availability of the data used to support the conclusions of your study, including accession codes to public repositories, references to source data that may be published alongside the paper, unique identifiers such as URLs to data repository entries, or data set DOIs, and any other statement about data availability. At a minimum, you should include the following statement: “The data that support the findings of this study are available from the corresponding author upon request”, describing which data is available upon request and mentioning any restrictions on availability. If DOIs are provided, please include these in the Reference list (authors, title, publisher (repository name), identifier, year). For more guidance on how to write this section please see: <http://www.nature.com/authors/policies/data/data-availability-statements-data-citations.pdf>

CODE AVAILABILITY

Please include a “Code Availability” subsection in the Online Methods which details how your custom code is made available. Only in rare cases (where code is not central to the main conclusions of the paper) is the statement “available upon request” allowed (and reasons should be specified).

MATERIALS AVAILABILITY

ORCID

Nature Methods is committed to improving transparency in authorship. As part of our efforts in this direction, we are now requesting that all authors identified as ‘corresponding author’ on published papers create and link their Open Researcher and Contributor Identifier (ORCID) with their account on the Manuscript Tracking System (MTS), prior to acceptance. This applies to primary research papers only. ORCID helps the scientific community achieve unambiguous attribution of all scholarly contributions. You can create and link your ORCID from the home page of the MTS by clicking on ‘Modify my Springer Nature account’. For more information please visit www.springernature.com/orcid.

Sincerely,
Arunima

Arunima Singh, Ph.D.
Senior Editor
Nature Methods

Reviewers' Comments:

Reviewer #1:

Remarks to the Author:

Rice et al. present a novel framework that allows particle picking from tomograms without the need for user-provided templates or user-trained pickers. To date, most particle picking in cryo-ET involves template-based picking, which 1) requires knowing what you are looking for and 2) suffers from bias. To address this limitation, Rice et al. leverage a new machine learning approach, 'deep metric learning' for training on simulated tomogram datasets to develop a general model for particle picking. Importantly, they place the software into a versatile platform - Napari - in addition to providing documentation for how to use their software.

Overall, the paper represents an important step forward for cryo-ET given that this is the first general particle picker in cryo-ET. I have only minor comments:

Benchmarking TomoTwin on experimental cryo-ET data for in vitro samples (Fig. 3): This benchmark shows that the picked particles are indeed what the authors say they are (e.g., apoferritin), but it does not show the reader the overall accuracy of the approach. I appreciate that there is not a benchmark for this type of analysis in the field, but it also would help the paper to know the ability of TomoTwin on a crowded in vitro tomogram. Since this is the 'best case' scenario for contrast, the authors should show that TomoTwin can work optimally to find nearly every particle and to classify them correctly.

Rubisco reconstruction: The authors show convincingly that TomoTwin easily picks Rubisco particles from EMPIAR 10694. Can they show a comparison with the previous sub tomogram averaged Rubisco

structure EMD-3694 from the same paper? This will help to show the reader map quality and comparison with the original Rubisco publication.

Reviewer #2:

Remarks to the Author:

The manuscript proposed a deep-learning-based method to facilitate particle picking in cryoET tomograms. The authors used a deep metric learning method that maps the 3D particle features to a high-dimensional space and uses a triplet loss to constrain the distance among different features. The method is implemented in a new software named TomoTwin which provides two interfaces for the picking, a reference-based and a clustering-based one. While the method itself is interesting and new in the field of cryoEM, the performance seems not as good as expected. Some more tests might be still needed to demonstrate the performance of the method. The extremely low signal-to-noise ratio, the missing wedge effect and complicated cellular and experimental conditions are not well considered, further algorithm optimization might be still needed.

Concerns:

- 1) The missing wedge is always a big problem in cryoET particle picking, which elongates the particles and makes the adjacent particles merge and indistinguishable. The cellular contents are often crowded. The influence from the missing wedge should be considered, such as picking accuracy in the cellular tomograms with dense proteins, missed particles, and especially the accuracy of the z coordinate (assuming the lamella is located in the xy plane).
- 2) Is it possible to estimate how many rubisco particles are missed and mis-picked by TomoTwin in figure 4? Both recall and precision are important for estimating the performance of particle picking. At least, a 3D or series of sectioning views should be shown to demonstrate the picked particles in the map. A similar demonstration should also be useful for other tests in the manuscript. While the ground truth is missing, such figure can help users to estimate how many expected particles are missing.
- 3) in Fig 3b, it seems that only a small part of the particles appearing in the figure are picked. How many particles are missed? What do these missing particles look like?
- 4) if running both the reference-based and the cluster-based method on the tomogram of figure 4 and 5, how many particles can be picked? What is the difference between the results of different methods?
- 5) The clustering might require a large number of similar particles to form a distinguished center in high-dimensional space. If the number of target proteins is very small in a tomogram, can such a protein be picked by the clustered-based method or reference-based particles?

6) For the tomogram shown in Fig. 6, only the big and high-contrast particles are picked, and many particles are missed. Nearly all the dense small proteins appearing in the background of this tomogram are ignored. The initial training used proteins as small as 30 kD, but it seems quite few small particles are picked. These should be unexpected according to the designed performance. How to control which proteins will be picked or not, in case of too many proteins appear. What is the lower bound of molecule weight to be picked correctly?

7) An issue is that the training is based on a general model. Did the training consider the missing wedge? Will the variances of data collection parameters, such as angle step size, tilting angle range and noise level (total dose) in different tomograms, will influence the picking accuracy and recall?

8) If the tomogram contains many membrane structures, how do the membranes influence the picking? Can the membrane be considered as some kind of feature? how is the performance on picking membrane proteins and fiber-like proteins?

Reviewer #3:

Remarks to the Author:

The authors introduce a neural network for the identification of macromolecules in cellular tomograms. They have trained their network with a large variety of different macromolecules. The structure of the network is sensible, and the choice for deep-learning a metric is very good. The experiments show the validity of their approach with simulated as well as experimental data. Overall, this is a well-written article, by a group expert in electron tomography and the development of deep learning tools for solving image processing problems in this domain.

This reviewer suggests the publication of this manuscript unaltered.

Author Rebuttal, first revision:

Reviewers' Comments:

We thank the reviewers for their positive and constructive feedback, which aided us to further improve the manuscript. Below we include our detailed response to each point raised.

Reviewer #1:

Remarks to the Author:

Rice et al. present a novel framework that allows particle picking from tomograms without the need for user-provided templates or user-trained pickers. To date, most particle picking in cryo-ET involves template-based picking, which 1) requires knowing what you are looking for and 2) suffers from bias. To address this limitation, Rice et al. leverage a new machine learning approach, ‘deep metric learning’ for training on simulated tomogram datasets to develop a general model for particle picking. Importantly, they place the software into a versatile platform - Napari - in addition to providing documentation for how to use their software.

Overall, the paper represents an important step forward for cryo-ET given that this is the first general particle picker in cryo-ET. I have only minor comments:

We thank this reviewer for their very positive feedback.

Benchmarking TomoTwin on experimental cryo-ET data for in vitro samples (Fig. 3): This benchmark shows that the picked particles are indeed what the authors say they are (e.g., apoferritin), but it does not show the reader the overall accuracy of the approach. I appreciate that there is not a benchmark for this type of analysis in the field, but it also would help the paper to know the ability of TomoTwin on a crowded in vitro tomogram. Since this is the ‘best case’ scenario for contrast, the authors should show that TomoTwin can work optimally to find nearly every particle and to classify them correctly.

As pointed out by this reviewer, it is difficult to quantify precision and recall of a picking program when using experimental tomograms. This challenge has been one of the main reasons behind our extensive experimentation with simulated tomograms wherein we can precisely measure true/false positive/negatives and report them in the F1 picking score.

Nevertheless, we estimated precision and recall for the larger proteins TcdA1 and apoferritin as follows. We manually evaluated the picking for one tomogram by counting true-positive (picked) and false-negative (missed) particles (Supplementary Fig. 1). For TcdA1 9 of 11 particles (recall: 0.81) and for apoferritin 126 of 137 (recall: 0.91) were picked. For both proteins, no false positive selections were found (precision 1.0). We think that this is a valid indication that TomoTwin picked almost all particles of TcdA1 and apoferritin in an accurate manner. However, for small particles, such as RhsA, that are difficult to distinguish from partially denatured or broken particles by eye in experimental tomograms, it is infeasible to measure the precision and recall manually. We therefore estimated the precision using the 2D classification procedure as described now in more detail in the main text (page 12) and in the methods (page 29).

To allow the reader a better visual inspection of our results, we also prepared a video going through the tomogram in z direction with the individual picks for TcdA1, apoferritin and RhsA highlighted (Supplementary Video 1).

Rubisco reconstruction: The authors show convincingly that TomoTwin easily picks Rubisco particles from EMPIAR 10694. Can they show a comparison with the previous sub tomogram averaged Rubisco structure EMD-3694 from the same paper? This will help to show the reader map quality and comparison with the original Rubisco publication.

This is an excellent suggestion. We have added a comparison with the previous Rubisco structure (EMD-3694) (Supplementary Fig. 11) in the revised manuscript. The resolution and quality of our reconstruction is superior to the previous Rubisco structure as can be seen from the comparison. However, since we cannot rule out that this is also partly due to the different workflow we used to process the data, we are more conservative with our statement in the main text:

"Comparing the map of RuBisCO from TomoTwin raw picked particles to the original (EMDB-3694) shows that the reconstruction from TomoTwin picked particles is of a comparable if not superior quality and thus TomoTwin picking does not limit the resolution that can be achieved by subtomogram averaging."

Reviewer #2:

Remarks to the Author:

The manuscript proposed a deep-learning-based method to facilitate particle picking in cryoET tomograms. The authors used a deep metric learning method that maps the 3D particle features to a high-dimensional space and uses a triplet loss to constrain the distance among different features. The method is implemented in a new software named TomoTwin which provides two interfaces for the picking, a reference-based and a clustering-based one. While the method itself is interesting and new in the field of cryoEM, the performance seems not as good as expected. Some more tests might be still needed to demonstrate the performance of the method. The extremely low signal-to-noise ratio, the missing wedge effect and complicated cellular and experimental conditions are not well considered, further algorithm optimization might be still needed.

Concerns:

1) The missing wedge is always a big problem in cryoET particle picking, which elongates the particles and makes the adjacent particles merge and indistinguishable. The cellular contents are often crowded. The influence from the missing wedge should be considered, such as picking accuracy in the cellular tomograms with dense proteins, missed particles, and especially the accuracy of the z coordinate (assuming the lamella is located in the xy plane).

Missing wedge:

We agree that the missing wedge is a problem in cryo-ET and have gone to great lengths to address it in several aspects of the algorithm development. We have addressed this reviewer's concerns

regarding the consequences of the missing wedge artefacts in detail in response to comment #7 (see below).

Picking accuracy in cellular tomograms with dense proteins:

In order to further explore the ability of TomoTwin to pick particles in crowded tomograms, we performed an additional experiment wherein we simulated a version of the generalization tomogram with a 5-fold increase in the number of particles and measured the picking accuracy with TomoTwin. The new figure is included as Supplementary Fig. 5 and discussed in the results section (page 11):

“In order to quantitatively measure the effect of increased particle density on picking quality, a version of the generalization tomogram was simulated in which the number of particles per protein in the tomogram was increased by 5-fold in order to replicate a highly crowded environment (Supplementary Fig. 5). In this densely packed tomogram, we observe an overall mean F1 picking score of 0.82 indicating that while particle picking in dense environments containing many proteins poses an additional challenge, the picking performance of the TomoTwin general model remains unequivocal.”

Picking accuracy in cellular tomograms regarding missed particles:

Since this point depends very much on the specific data set used, we address it specifically in our answers to comments #2, #3, and #6 (see below).

Picking accuracy in cellular tomograms regarding the z coordinate:

TomoTwin uses a 3D convolutional neural network which analyzes each subvolume of the tomogram as a whole 3D object rather than weighing the Z axis separately as would a network that analyzes 3D objects as a series of 2D slices similar to the how humans commonly visualize tomograms. Therefore, the accuracy of locating particles in Z will be similar to the accuracy in X,Y. We have made this more clear in the manuscript by adding the new Supplementary Figure 13a which shows an X,Z view of picked particles in the simulated generalization tomogram. Although particles were picked on the raw tomogram in all cases, we show a denoised tomogram here for visualization purposes as this allows the reader to clearly see the Z coordinate is accurate for even small proteins picked with TomoTwin. Supplementary Figure 13b-e shows analogous views for the other tomograms.

2) Is it possible to estimate how many rubisco particles are missed and mis-picked by TomoTwin in figure 4? Both recall and precision are important for estimating the performance of particle picking. At least, a 3D or series of sectioning views should be shown to demonstrate the picked particles in the map. A similar demonstration should also be useful for other tests in the manuscript. While the ground truth is missing, such figure can help users to estimate how many expected particles are missing.

To address this point, we manually counted the selected and missed particles for a reference region to estimate the performance of particle picking (Supplementary Fig. 10a-c). We furthermore followed the excellent suggestion and created a video that slowly moves through the tomogram in z direction and highlights the individual picks (Supplementary Video 2).

3) in Fig 3b, it seems that only a small part of the particles appearing in the figure are picked. How many particles are missed? What do these missing particles look like?

We estimated precision and recall for the larger complexes TcdA1 and apoferritin as follows. We manually evaluated the picking for one tomogram by counting true-positive (picked) and false-negative (missed) particles (Supplementary Fig. 1). For TcdA1 9 of 11 particles (recall: 0.81) and for apoferritin 126 of 137 (recall: 0.91) were picked. For both proteins, no false positive selections were found (precision 1.0). We think that this is a valid indication that TomoTwin picked almost all particles of TcdA1 and apoferritin in an accurate manner.

We reevaluated the picking for RhsA and found that using a lower picking threshold which included more particles still yielded good 2D classes indicating that many previously unpicked particles are RhsA which are now picked. We updated Figure 3 accordingly. However, for small particles, such as RhsA, that are difficult to distinguish from partially denatured or broken particles by eye in experimental tomograms, it is infeasible to measure the precision and recall manually. We therefore estimated the precision using the 2D classification procedure as described now in more detail in the main text (page 12) and in the methods (page 29).

To allow the reader a better visual inspection of our results, we also prepared a video going through the tomogram in z direction with the individual picks for TcdA1, apoferritin and RhsA highlighted (Supplementary Video 1).

4) if running both the reference-based and the cluster-based method on the tomogram of figure 4 and 5, how many particles can be picked? What is the difference between the results of different methods?

Thanks for the excellent suggestion. In Supplementary Fig. 10a,c one can see that most picks are identical between both procedures when applied to the RuBisCO tomogram. Only few particles are uniquely picked by either the reference-based or cluster-based workflow.

For the simulated generalization tomogram in Figure 5, we report the F1 score for each protein when picked with the reference-based and clustering-based workflows (Fig. 5e). Supplementary Table 2 now shows the false positive, false negative, true positives and additionally the precision and recall for both workflows on this tomogram. As the similar f1 score indicate, the picking of the reference and clustering workflow is interchangeable in most cases. However, in the case of

1FPY the reference-based picking gave an F1 score of 0.69 whereas the cluster-based workflow gave an F1 score of 0.96, which is a notable difference. This is due to the fact that not all references work equally well. In comparison, for Supplementary Figure 12 (see our response to comment #7), we selected 5 random references and reported the best results to compensate for a possible reference dependency. In this case, we reached a F1 score 0.95 for 1FPY with the reference workflow. Therefore, in the online tutorial, we suggest picking a few references for each protein if possible and comparing their results. The reference dependency is additionally mentioned in the manuscript in line 330 – 332.

For the *M. pneumoniae* dataset, we show in white the picks that can be attained with both the reference-based and clustering workflow and in blue the picks that were only attained from the reference-based workflow. In total there were 325 particles picked with the reference-based and 306 particles picked with the clustering workflow, again demonstrating that the picking results are very similar. Furthermore, we manually counted for the reference results 299 true positive picks, 6 false positive picks and 25 false negative picks which results in a precision of 0.98 and a recall of 0.92.

5) The clustering might require a large number of similar particles to form a distinguished center in high-dimensional space. If the number of target proteins is very small in a tomogram, can such a protein be picked by the clustered-based method or reference-based particles?

This is an excellent observation and indeed the clustering-based workflow depends on how well defined a cluster is in the UMAP which in turn depends on the number of similar particles in the tomogram. We added the following sentence in page 17 of the manuscript to address this issue:

“However, it does require that the abundance of the protein is high enough to form a cluster.”

This limitation additionally highlights the advantage of having two workflows, the reference-based and clustering workflow to complement each other and address the diversity of samples studied with cryo-ET.

6) For the tomogram shown in Fig. 6, only the big and high-contrast particles are picked, and many particles are missed. Nearly all the dense small proteins appearing in the background of this tomogram are ignored.

In the tomogram in Figure 6, we purposefully only show the picking of the two clusters which we have processed by subtomogram averaging in Figure 6d. The dense, small proteins appearing in the background of the tomogram are difficult to distinguish from one another by eye at this resolution. While they can be picked with the clustering workflow, their small size in combination with the 10 Å pixel size used, precludes us from identifying which protein they are by

subtomogram averaging. It is therefore impossible to evaluate the accuracy of TomoTwin in picking them and we excluded them from the analysis for this manuscript.

The initial training used proteins as small as 30 kD, but it seems quite few small particles are picked. These should be unexpected according to the designed performance.

The small proteins are picked, however, due to the reasons described above, we did not include them in this analysis. Upon generalization, we also found that the lower limit of accuracy was about 150 kDa, even though the training data contained proteins as small as 30 kDa, because TomoTwin operates on downsampled tomograms ($\sim 10 \text{ \AA}/\text{pix}$).

How to control which proteins will be picked or not, in case of too many proteins appear.

For each tomogram, users can pick references for each of the proteins they wish to pick, or identify as many clusters in the clustering workflow as they wish to pick. On a per-protein basis the number of particles picked is entirely up to the user as in the TomoTwin plugin in Napari, users can threshold the picks based on network confidence as well as size. If too many particles are picked and users are worried there are false positive picks, one can simply increase the minimum similarity threshold and the adjusted picking will be shown on the fly. Similarly, users can use the size thresholding to filter out unwanted picks based on particle size. Supplementary Fig. 2 shows a snapshot of this process in Napari.

What is the lower bound of molecule weight to be picked correctly?

Thank you for this very important point about the lower bound of molecular weight to be picked correctly as it is an important point for the manuscript which we had simply overlooked. We have now added a section to the discussion to be sure this point is clear to readers:

“Finally, although the training data included proteins as small as 30 kDa, during generalization the expected lower limit of accurate picking is approximately 150 kDa on account of TomoTwin picking on downsampled tomograms ($\sim 10 \text{ \AA}/\text{pix}$). While particles smaller than this size can potentially be located in experimental tomograms, evaluating the accuracy of such picks becomes intractable through subtomogram averaging.”

7) An issue is that the training is based on a general model. Did the training consider the missing wedge? Will the variances of data collection parameters, such as angle step size, tilting angle range and noise level (total dose) in different tomograms, will influence the picking accuracy and recall?

The general model created with deep metric learning is in fact the primary novel advancement of TomoTwin. However, we thank this reviewer for pointing out that it could be a problem if important experimental parameters such as the missing wedge were not taken into account during model training.

The training of TomoTwin considers the missing wedge in multiple regards. First, all training, validation, and generalization tomograms were reconstructed from tilt series simulated from -60° to 60° therefore with a 60° missing wedge (page 23). In addition, during data augmentation the random rotation parameters were chosen such that the missing wedge is oriented appropriately (page 26), an idea first presented in Moebel et al., 2021 when describing the training of DeepFinder.

In order to test the variances of data collection parameters on the picking, we performed additional experiments wherein we simulated generalization tomograms with a variety of tilt ranges and total doses (new Supplementary Fig. 12). To control for possible bias from the reference particle used for picking, five reference particles were used for each protein and the reference that returned the most consistent picking result across the parameters was reported. Overall, we observe a 5.4 % decrease in mean F1 picking performance when the tilt range is restricted from $-60^\circ - 60^\circ$ to $-50^\circ - 50^\circ$. It decreases to 10.3 % when the tilt range is reduced to $-40^\circ - 40^\circ$ (Supplementary Fig. 12a).

Decreasing the tilt range increases the effects of missing wedge artifact on the reconstructed tomogram resulting in a stronger deformation of each reconstructed particle. This effect is particularly pronounced on long, thin particles such as carbon monoxide dehydrogenase (PDB code: 1OAO) making accurate particle picking significantly more challenging. Similarly, decreasing the total electron dose directly reduces the protein signal resulting in a tomogram with a reduced signal to noise ratio. Examining this, we observe a 2.3 % and 8.6 % decrease in F1 picking performance when the total dose is restricted from $150 \text{ e}/\text{\AA}^2$ to $135 \text{ e}/\text{\AA}^2$ and $120 \text{ e}/\text{\AA}^2$, respectively (Supplementary Fig. 12b).

In addition, the datasets used to evaluate the performance of TomoTwin all vary from each other in at least one data collection parameter, in some cases widely. We have gathered the data collection parameters for each dataset analyzed in this manuscript and used them to create Supplementary Table 1 to highlight to readers that TomoTwin has been tested on experimental data from a variety of experimental setups.

Overall, while a decrease in picking performance with the increase in corruption or lack of protein signal in tomograms due to missing wedge or a decreased total dose is expected, TomoTwin leverages the high degree of adaptability rooted in its deep metric learning backbone to achieve accurate particle picking.

We have added a completely new chapter and the new figures to the revised manuscript.

8) If the tomogram contains many membrane structures, how do the membranes influence the picking?

To evaluate the influence of membrane structures, we conducted a new experiment. New versions of the generalization tomogram were simulated with an increasing number of vesicles (Rebuttal Figure 1a-c). We then evaluated the F1 score for each of the protein in these additional tomograms (Rebuttal Figure 1d). The average F1 score was 0.90, 0.88 and 0.86 for the tomogram with 4, 8 and 12 vesicles respectively. Thus, based on this data, there is no indication that membrane structures in tomograms have a strong influence on the picking of soluble proteins.

Can the membrane be considered as some kind of feature?

In principle, membranes could be selected as their own class. However, this is aggravated by the different appearance of natural membranes such as thickness, curvature and density due to different lipid composition and membrane protein content.

How is the performance on picking membrane proteins and fiber-like proteins?

TomoTwin has so far not been trained to pick membrane proteins or filamentous proteins as we have clearly described in the discussion “Currently, the model is not designed to pick membrane proteins or filaments”. We are working on these tasks, however, these are very challenging tasks as described below and therefore beyond the scope of this manuscript.

1. The training of TomoTwin was made possible by existing algorithms for the accurate simulation of TEM images containing soluble proteins. In order to develop a model to pick membrane proteins, we would need to generate simulated training data containing membrane proteins inside membranes. No program exists at the moment to create such TEM images of membrane proteins.
2. The simulator does not yet produce naturally positioned and bent filaments. Thus, it is difficult to train TomoTwin appropriately. Additionally, we would have to implement an adapted filament tracing.

Reviewer #3:

Remarks to the Author:

The authors introduce a neural network for the identification of macromolecules in cellular tomograms. They have trained their network with a large variety of different macromolecules.

The structure of the network is sensible, and the choice for deep-learning a metric is very good. The experiments show the validity of their approach with simulated as well as experimental data. Overall, this is a well-written article, by a group expert in electron tomography and the development of deep learning tools for solving image processing problems in this domain.

This reviewer suggests the publication of this manuscript unaltered.

We thank this reviewer for their very positive feedback.

Rebuttal Fig. 1: Influence of membranes on TomoTwin. **a-c**, To evaluate the influence of membrane structures on TomoTwin, generalization tomograms with 4 vesicles (**a**), 8 vesicles (**b**) and 12 vesicles (**c**) were simulated and reconstructed. **d**, For each tomogram the F1 score was calculated using the reference-based workflow. The average F1 score was 0.90, 0.88 and 0.86 for the tomogram with 4, 8 and 12 vesicles, respectively.

Decision Letter, second revision:

Dear Stefan,

Thank you for submitting your revised manuscript "TomoTwin: Generalized 3D Localization of Macromolecules in Cryo-electron Tomograms with Structural Data Mining" (NMETH-A49654B). It has now been seen by the original referees and their comments are below. The reviewers find that the

paper has improved in revision, and therefore we'll be happy in principle to publish it in Nature Methods, pending minor revisions to comply with our editorial and formatting guidelines.

TRANSPARENT PEER REVIEW

Nature Methods offers a transparent peer review option for new original research manuscripts submitted from 17th February 2021. We encourage increased transparency in peer review by publishing the reviewer comments, author rebuttal letters and editorial decision letters if the authors agree. Such peer review material is made available as a supplementary peer review file. Please state in the cover letter 'I wish to participate in transparent peer review' if you want to opt in, or 'I do not wish to participate in transparent peer review' if you don't. Failure to state your preference will result in delays in accepting your manuscript for publication.

ORCID

Sincerely,
Arunima

Arunima Singh, Ph.D.
Senior Editor

Nature Methods

Reviewer #1 (Remarks to the Author):

The authors have addressed my comments.

Reviewer #2 (Remarks to the Author):

All comments have been well addressed.

Final Decision Letter:

Dear Stefan,

I am pleased to inform you that your Article, "TomoTwin: Generalized 3D Localization of Macromolecules in Cryo-electron Tomograms with Structural Data Mining", has now been accepted for publication in Nature Methods. Your paper is tentatively scheduled for publication in our June print issue, and will be published online prior to that. The received and accepted dates will be June 24, 2022 and April 12, 2023. This note is intended to let you know what to expect from us over the next month or so, and to let you know where to address any further questions.

Once your paper is typeset, you will receive an email with a link to choose the appropriate publishing options for your paper and our Author Services team will be in touch regarding any additional information that may be required.

Please note that *Nature Methods* is a Transformative Journal (TJ). Authors may publish their research with us through the traditional subscription access route or make their paper immediately open access through payment of an article-processing charge (APC). Authors will not be required to make a final decision about access to their article until it has been accepted. ](https://www.springernature.com/gp/open-research/transformative-journals) Find out more about Transformative Journals

Authors may need to take specific actions to achieve compliance with funder and institutional open access mandates. If your research is supported by a funder that requires immediate open access (e.g. according to Plan S principles) then you should select the gold OA route, and we will direct you to the compliant route where possible. For authors selecting the subscription publication route, the journal's standard licensing terms will need to be accepted, including self-archiving policies. Those licensing terms will supersede any other terms that the author or any third party may assert apply to any version of the manuscript.

Your paper will now be copyedited to ensure that it conforms to Nature Methods style. Once proofs are generated, they will be sent to you electronically and you will be asked to send a corrected version within 24 hours. It is extremely important that you let us know now whether you will be difficult to contact over the next month. If this is the case, we ask that you send us the contact information (email, phone and fax) of someone who will be able to check the proofs and deal with any last-minute problems.

If, when you receive your proof, you cannot meet the deadline, please inform us at rjsproduction@springernature.com immediately.

Once your manuscript is typeset and you have completed the appropriate grant of rights, you will receive a link to your electronic proof via email with a request to make any corrections within 48 hours. If, when you receive your proof, you cannot meet this deadline, please inform us at rjsproduction@springernature.com immediately.

Once your paper has been scheduled for online publication, the Nature press office will be in touch to confirm the details.

Once your paper has been scheduled for online publication, the Nature press office will be in touch to confirm the details.

Content is published online weekly on Mondays and Thursdays, and the embargo is set at 16:00 London time (GMT)/11:00 am US Eastern time (EST) on the day of publication. If you need to know the exact publication date or when the news embargo will be lifted, please contact our press office after you have submitted your proof corrections. Now is the time to inform your Public Relations or Press Office about your paper, as they might be interested in promoting its publication. This will allow them time to prepare an accurate and satisfactory press release. Include your manuscript tracking number NMETH-A49654C and the name of the journal, which they will need when they contact our office.

About one week before your paper is published online, we shall be distributing a press release to news organizations worldwide, which may include details of your work. We are happy for your institution or funding agency to prepare its own press release, but it must mention the embargo date and Nature Methods. Our Press Office will contact you closer to the time of publication, but if you or your Press Office have any inquiries in the meantime, please contact press@nature.com.

Nature Portfolio journals [encourage authors to share their step-by-step experimental protocols](https://www.nature.com/nature-research/editorial-policies/reporting-standards#protocols) on a protocol sharing platform of their choice. Nature Portfolio's Protocol Exchange is a free-to-use and open resource for protocols; protocols deposited in Protocol Exchange are citable and can be linked from the published article. More details can be found at www.nature.com/protocolexchange/about.

Please note that you and any of your coauthors will be able to order reprints and single copies of the issue containing your article through Nature Portfolio 's reprint website, which is located at <http://www.nature.com/reprints/author-reprints.html>. If there are any questions about reprints please send an email to author-reprints@nature.com and someone will assist you.

Best regards,
Arunima

Arunima Singh, Ph.D.
Senior Editor
Nature Methods